# Internal Cross-layer Gradients for Extending Homogeneity to Heterogeneity in Federated Learning

**Yun-Hin Chan, Rui Zhou, Running Zhao, Zhihan Jiang & Edith C.H. Ngai**[*]
Department of Electrical and Electronic Engineering, The University of Hong Kong
`{chanyunhin,zackery,rnzhao,zhjiang}@connect.hku.hk,`
`chngai@eee.hku.hk`

## Abstract

Federated learning (FL) inevitably confronts the challenge of system heterogeneity in practical scenarios. To enhance the capabilities of most model-homogeneous FL methods in handling system heterogeneity, we propose a training scheme that can extend their capabilities to cope with this challenge. In this paper, we commence our study with a detailed exploration of homogeneous and heterogeneous FL settings and discover three key observations: (1) a positive correlation between client performance and layer similarities, (2) higher similarities in the shallow layers in contrast to the deep layers, and (3) the smoother gradient distributions indicate the higher layer similarities. Building upon these observations, we propose InCo Aggregation that leverages internal cross-layer gradients, a mixture of gradients from shallow and deep layers within a server model, to augment the similarity in the deep layers without requiring additional communication between clients. Furthermore, our methods can be tailored to accommodate model-homogeneous FL methods such as FedAvg, FedProx, FedNova, Scaffold, and MOON, to expand their capabilities to handle the system heterogeneity. Copious experimental results validate the effectiveness of InCo Aggregation, spotlighting internal cross-layer gradients as a promising avenue to enhance the performance in heterogeneous FL.

## 1 Introduction

Federated learning (FL) is proposed to enable a federation of clients to effectively cooperate towards a global objective without exchanging raw data (McMahan et al., 2017). While FL makes it possible to fuse knowledge in a federation with privacy guarantees (Huang et al., 2021; McMahan et al., 2017; Jeong & Hwang, 2022), its inherent attribute of system heterogeneity (Li et al., 2020a), i.e., varying resource constraints of local clients, may hinder the training process and even lower the quality of the jointly-learned models (Kairouz et al., 2021; Li et al., 2020a; Mohri et al., 2019; Gao et al., 2022).

System heterogeneity refers to a set of varying physical resources $\{R_i\}_{i=1}^n$, where $R_i$ denotes the resource of client $i$, a high-level idea of resource that holistically governs the aspects of computation, communication, and storage. Existing works cater to system heterogeneity through a methodology called model heterogeneity, which aligns the local models of varying architectures to make full use of local resources (Diao et al., 2021; Baek et al., 2022; Alam et al., 2022; Huang et al., 2022; Fang & Ye, 2022; Lin et al., 2020). Specifically, model heterogeneity refers to a set of different local models $\{M_i\}_{i=1}^n$ with $M_i$ being the model of client $i$. Let $R(M)$ denote the resource requirement for the model $M$. Model heterogeneity is a methodology that manages to meet the constraints $\{R(M_i) \leq R_i\}_{i=1}^n$. In the case of model heterogeneity, heterogeneous devices are allocated to a common model prototype tailored to their varying sizes, such as ResNets with different depths or widths of layers (Liu et al., 2022; Diao et al., 2021; Horvath et al., 2021; Baek et al., 2022; Caldas et al., 2018; Ilhan et al., 2023), strides of layers (Tan et al., 2022), or numbers of kernels (Alam et al., 2022), to account for their inherent resource constraints. While several methods have been proposed to incorporate heterogeneous models into federated learning (FL), their performances often fall short compared to FL training using homogeneous models of the same size (He et al., 2020; Diao et al., 2021). Therefore, gaining a comprehensive understanding of the factors that limit the performance of heterogeneous models in FL is imperative. The primary objective of this paper is to investigate

---

[*]Corresponding author.

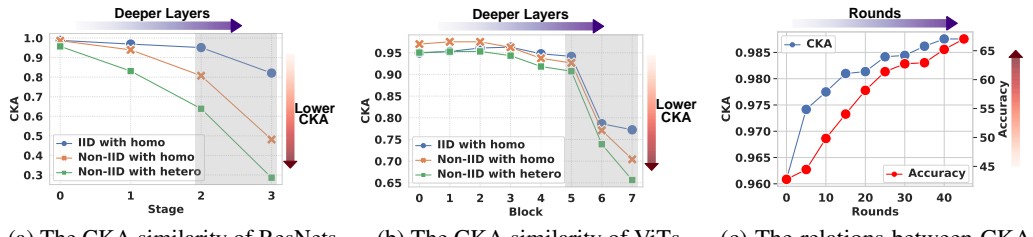

Figure 1: CKA similarity in different environments and the relation between accuracy and CKA similarity. (a) and (b): The CKA similarity of different federated settings. (c): The positive relation between CKA and accuracy during the training process.

the underlying reasons behind this limitation and propose a potential solution that acts as a bridge between model homogeneity and heterogeneity to tackle this challenge.

In light of this, we first conduct a case study to reveal the obstacles affecting the performance of heterogeneous models in FL. The observations from this case study are enlightening: (1) With increasing heterogeneity in data distributions and model architectures, we observe a decline in model accuracy and layer-wise similarity (layer similarity) as measured by Centered Kernel Alignment (CKA)[1] (Kornblith et al., 2019), a quantitative metric of bias (Luo et al., 2021; Raghu et al., 2021); (2) The deeper layers share lower layer similarity across the clients, while the shallower layers exhibit greater alignment. These insights further shed light on the notion that shallow layers possess the ability to capture shared features across diverse clients, even within the heterogeneous FL setting. Moreover, these observations indicate that the inferior performances in heterogeneous FL are related to the lower similarity in the deeper layers. Motivated by these findings, we come up with an idea: **Can we enhance the similarity of deeper layers, thereby attaining improved performance?**

To answer this question, we narrow our focus to the gradients, as the dissimilarity of deep layers across clients is a direct result of gradient updates (Ruder, 2016; Chen et al., 2021). Interestingly, we observe that (3) the gradient distributions originating from shallow layers are smoother and possess higher similarity than those from deep layers, establishing a connection between the gradients and the layer similarity. Therefore, inspired by these insights, we propose a method called **InCo Aggregation**, deploying different model splitting methods and utilizing the **In**ternal **Cro**ss-layer gradients (**InCo**) in a server model to improve the similarity of its deeper layers without additional communications with the clients. More specifically, cross-layer gradients are mixtures of the gradients from the shallow and the deep layers. We utilize cross-layer gradients as internal knowledge, effectively transferring knowledge from the shallow layers to the deep layers. Nevertheless, mixing these gradients directly poses a significant challenge called gradient divergence (Wang et al., 2020; Zhao et al., 2018). To tackle this issue, we normalize the cross-layer gradients and formulate a convex optimization problem that rectifies their directions. In this way, InCo Aggregation automatically assigns optimal weights to the cross-layer gradients, thus avoiding labor-intensive parameter tuning. Furthermore, **InCo Aggregation can extend to model-homogeneous FL methods that previously do not support model heterogeneity**, such as FedAvg(McMahan et al., 2017), FedProx (Li et al., 2020b), FedNova (Wang et al., 2020), Scaffold (Karimireddy et al., 2020), and MOON (Li et al., 2021a), to develop their abilities in managing the model heterogeneity problem.

Our main contributions are summarized as follows:

- We first conduct a case study on homogeneous and heterogeneous FL settings and find that (1) client performance is positively correlated to layer similarities across different client models, (2) similarities in the shallow layers are higher than the deep layers, and (3) smoother gradient distributions hint for higher layer similarities.

- We propose InCo Aggregation, applying model splitting and the internal cross-layer gradients inside a server model. Moreover, our methods can be seamlessly applied to various model-homogeneous FL methods, equipping them with the ability to handle model heterogeneity.

- We establish the non-convex convergence of utilizing cross-layer gradients in FL and derive the convergence rate.

- Extensive experiments validate the effectiveness of InCo Aggregation, showcasing its efficacy in strengthening model-homogeneous FL methods for heterogeneous FL scenarios.

---

[1]The detailed descriptions for CKA are introduced in Appendix A.

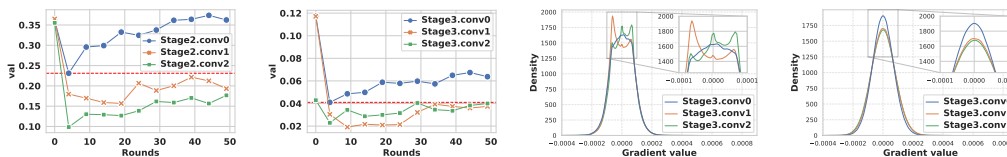

(a) Similarity of gradients in Stage 2.

(b) Similarity of gradients in Stage 3.

(c) Gradient distributions in Non-IID with hetero.

(d) Gradient distributions in IID with homo.

Figure 2: Cross-environment similarity and gradients distributions. (a) and (b): Similarity from Stage 2 and Stage 3. (c) and (d): The gradient distributions of Non-IID with hetero and IID with homo.

## 2 PRELIMINARY

To investigate the performance of clients in diverse federated learning settings, we present a case study encompassing both homogeneous and heterogeneous model architectures with CIFAR-10 and split data based on IID and Non-IID with ResNets (He et al., 2016) and ViTs (Dosovitskiy et al., 2020). We use CKA (Kornblith et al., 2019) similarities among models to measure the level of bias exhibited by each model. More detailed results of the case study are provided in Appendix G.

### 2.1 A CASE STUDY IN DIFFERENT FEDERATED LEARNING ENVIRONMENTS

*Case Analysis.* Generally, we find three intriguing observations from Table 1 and Figure 1:
(i) The deeper layers or stages have lower CKA similarities than the shallow layers. (ii) The settings with higher accuracy also obtain higher CKA similarities in the deeper layers or stages. (iii) The CKA similarity is positively related to the accuracy of clients, as shown in Figure 1c. These observations indicate that increasing the similarity of deeper layers can serve as a viable approach to improving client performance. Considering that shallower layers exhibit higher similarity, a potential direction emerges: to improve the CKA similarity in deeper layers according to the knowledge from the shallower layers.

Table 1: Accuracy of the case study.

| | Settings | Test Accuracy |
|---|---|---|
| ResNet | IID with homo | 81.0 |
| | Non-IID with homo | 62.3($\downarrow$18.7) |
| | Non-IID with hetero | 52.3($\downarrow$28.7) |
| ViT | IID with homo | 81.0 |
| | Non-IID with homo | 54.8($\downarrow$26.2) |
| | Non-IID with hetero | 50.1($\downarrow$30.9) |

### 2.2 DEEP INSIGHTS OF GRADIENTS IN THE SHALLOWER LAYERS

*Gradients as Knowledge.* In FL, there are two primary types of knowledge that can be utilized: features, which are outputs from middle layers, and gradients from respective layers. We choose to use gradients as our primary knowledge for two essential reasons. Firstly, our FL environment lacks a shared dataset, impeding the establishment of a connection between different clients using features derived from the same data. Secondly, utilizing features in FL would significantly increase communication overheads. Hence, taking these practical considerations into account, we select gradients as the knowledge.

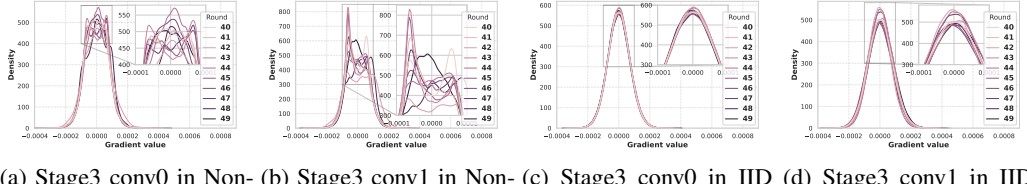

(a) Stage3 conv0 in Non-IID with hetero.

(b) Stage3 conv1 in Non-IID with hetero.

(c) Stage3 conv0 in IID with homo.

(d) Stage3 conv1 in IID with homo.

Figure 3: The gradient distributions from round 40 to 50 in different environments.

*Cross-environment Similarity.* In this subsection, we deeply investigate the cross-environment similarity of gradients between two environments, i.e., *IID with homo* and *Non-IID with hetero*, to shed light on the disparities between shallow and deep layers in the same stage[2] and identify the gaps between the homogeneous and heterogeneous FL. As depicted in Figure 2a and 2b, gradients from shallow layers (*Stage2.conv0* and *Stage3.conv0*) exhibit higher cross-environment CKA similarity than those from deep layers such as *Stage2.conv1*, and *Stage3.conv2*. Notably, even the lowest similarities (red lines) in *Stage2.conv0* and *Stage3.conv0* surpass the highest similarities in deep layers. These findings underscore the superior quality of gradients obtained from shallow layers

---

[2]We discuss a shallow layer (the first layer with the same shape in a stage) and deep layers (remaining layers) within a stage for ResNets and a block for ViTs. The gradient analyses for ViTs are introduced in Appendix G.3

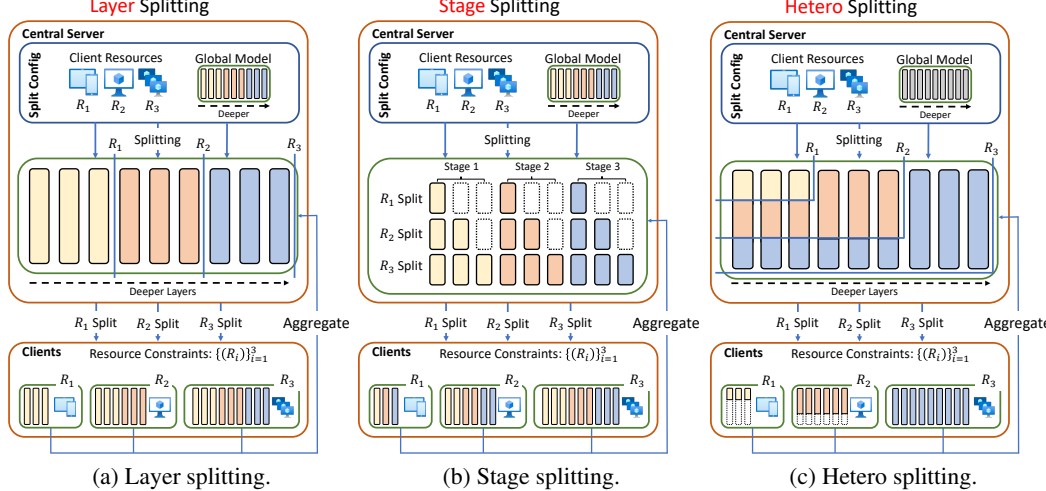

(a) Layer splitting.      (b) Stage splitting.      (c) Hetero splitting.

Figure 5: The system architecture of three different model splitting methods: (a) layer splitting, (b) stage splitting, and (c) heterogeneous (hetero) splitting. (a): Layer splitting divides the entire model layer by layer. (b): Stage splitting separates each stage layer by layer. (c): Hetero splitting partitions the whole model in different widths and depths depending on the available resources $R_i$ of client $i$.

relative to those obtained from deep layers, and also indicate that the layers within the same stage exhibit similar patterns to the layers throughout the entire model.

*Gradient Distributions.* To dig out the latent relations between gradients and layer similarity, we delve deeper into the analysis of gradient distributions across different FL environments. More specifically, the comparison of Figure 2c and Figure 2d reveals that gradients from shallow layers (*Stage3.conv0*) exhibit greater similarity in distribution between *Non-IID with hetero* and *IID with homo* environments, in contrast to deep layers (*Stage3.conv1* and *Stage3.conv2*). Additionally, as depicted in Figure 3c and Figure 3d, the distributions of gradients from a deep layer (Figure 3d) progressively approach the distribution of gradients from a shallow layer (Figure 3c) with each round, in contrast to Figure 3a and Figure 3b, where the distributions from deep layers (Figure 3b) are less smooth than those from shallow layers (Figure 3a) in *Non-IID with hetero* during rounds 40 to 50. Consequently, drawing from the aforementioned gradient analysis, we can enhance the quality of gradients from deep layers in *Non-IID with hetero* environments by leveraging gradients from shallow layers, i.e., cross-layer gradients as introduced in the subsequent section.

## 3   INCO AGGREGATION

We provide a concise overview of the three key components in InCo Aggregation at first. The first component is model splitting, including three types of model splitting methods, as shown in Figure 5. The second component involves the combination of gradients from a shallow layer and a deep layer, referred to as internal cross-layer gradients. To address gradient divergence, the third component employs gradient normalization and introduces a convex optimization formulation. We elaborate on these three critical components of InCo Aggregation as follows.

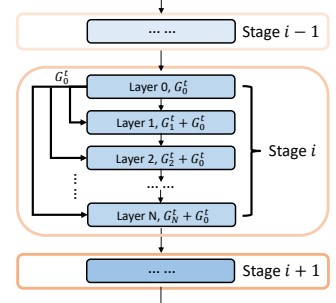

Figure 4: Cross-layer gradients for the server model in InCo.

### 3.1   MODEL SPLITTING

To facilitate model heterogeneity, we propose three model splitting methods: layer splitting, stage splitting, and hetero splitting, as illustrated in Figure 5. These methods distribute models with varying sizes to clients based on their available resources, denoted as $R_i$. In layer splitting, the central server initiates a global model and splits it layer by layer, considering the client resources $R_i$, as depicted in Figure 5a. In contrast, stage splitting separates each stage layer by layer in Figure 5b. For instance, Figure 5b illustrates how the smallest client with $R_1$ resources obtains the first layer from each stage in stage splitting, whereas it acquires the first three layers from the entire model in layer splitting. Furthermore, hetero splitting, depicted in Figure 5c, involves the server splitting the global model into distinct widths and depths for

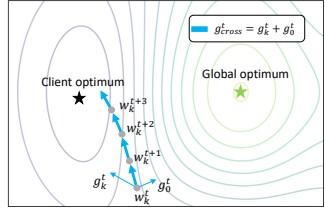 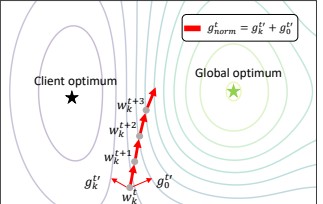 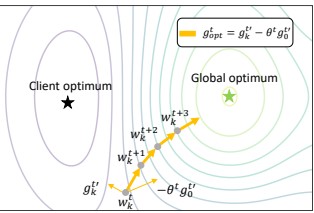

(a) Gradient divergence.      (b) Normalized gradients.      (c) Normalized and optimized.

Figure 6: A depiction of gradient divergence, as shown in Figure 6a, along with its solutions. Despite the normalization portrayed in Figure 6b, the impact of gradient divergence persists. To mitigate this issue, we propose a convex optimization problem that is restricting gradient directions, as demonstrated in Figure 6c and supported by Theorem 3.1.

different clients, similar to the approaches in HeteroFL (Diao et al., 2021) and FedRolex (Alam et al., 2022). Layer splitting and stage splitting offer flexibility for extending model-homogeneous methods to system heterogeneity, while hetero splitting enables the handling of client models with varied widths and depths. Finally, the server aggregates client weights based on their original positions in the server models.

### 3.2 INTERNAL CROSS-LAYER GRADIENTS

Deploying model splitting methods directly in FL leads to a significant decrease in client accuracy, as demonstrated in Table 1. However, based on the findings of the case study, we observe that gradients from shallow layers contribute to increasing the similarity among layers from different clients, and CKA similarity exhibits a positive correlation with client accuracy. Therefore, we enhance the quality of gradients from deep layers by incorporating the utilization of cross-layer gradients. More specifically, when a **server model** updates the deep layers, we combine and refine the gradients from these layers with the gradients from the shallower layers to obtain appropriately updated gradients. Figure 4 provides a visual representation of how cross-layer gradients are employed. We assume that this stage has $N$ layers. The first layer with the same shape in a stage (block) is referred to as **Layer 0**, and its corresponding gradients at time step $t$ are $G_0^t$. For Layer $k$, where $k \in \{1, 2, ..., N\}$ within the same stage, the cross-layer gradients are given by $G_k^t + G_0^t$. Despite a large number of works on short-cut paths in neural networks, our method differs fundamentally in terms of the goals and the operations. We provide a thorough discussion in Appendix B.

### 3.3 GRADIENTS DIVERGENCE ALLEVIATION

However, the direct utilization of cross-layer gradients leads to an acute issue known as gradient and weight divergence (Wang et al., 2020; Zhao et al., 2018), as depicted in Figure 6a. To counter this effect, we introduce gradient normalization (Figure 6b) and the proposed convex optimization problem to restrict gradient directions, as illustrated in Figure 6c.

*Cross-layer Gradients Normalization.* Figure 6b depicts the benefits of utilizing normalized gradients. The normalized cross-layer gradient $g_0^{t'} + g_k^{t'}$ directs the model closer to the global optimum than the original cross-layer gradient $g_0^t + g_k^t$. In particular, our normalization approach emphasizes the norm of gradients, i.e., $g_0^{t'} = g_0^t/||g_0^t||$ and $g_k^{t'} = g_k^t/||g_k^t||$. The normalized cross-layer gradient is computed as $(g_0^{t'} + g_k^{t'}) \times (||g_0^t|| + ||g_k^t||)/2$ in practice.

*Convex Optimization.* In addition to utilizing normalized gradients, incorporating novel projective gradients that leverage knowledge from both $g_0^t$ and $g_k^t$ serves to alleviate the detrimental impact of gradient divergence arising from the utilization of cross-layer gradients. Moreover, Our objective is to find the optimal projective gradients, denoted as $g_{opt}$, which strike a balance between being as close as possible to $g_k$ while maintaining alignment with $g_0$. This alignment ensures that $g_k$ is not hindered by the influence of $g_0$ while allowing $g_{opt}$ to acquire the beneficial knowledge for $g_k$ from $g_0$. In other words, we aim for $g_{opt}$ to capture the advantageous information contained within $g_0$ without impeding the progress of $g_k$. Pursuing this line of thought, we introduce a constraint aimed at ensuring the optimization directions of gradients, outlined as $\langle g_0^t, g_k^t \rangle \geq 0$, where $\langle \cdot, \cdot \rangle$ is the dot product. To establish a convex optimization problem incorporating this constraint, we denote the projected gradient as $g_{opt}$ and formulate the following primal convex optimization problem,

$$\min_{g_{opt}^t} \quad ||g_k^t - g_{opt}^t||_2^2, \quad s.t. \ \langle g_{opt}^t, g_0^t \rangle \geq 0, \tag{1}$$

where we preserve the optimization direction of $g_0^t$ in $g_{opt}^t$ while minimizing the distance between $g_{opt}^t$ and $g_k^t$. We prioritize the proximity of $g_{opt}^t$ to $g_k^t$ over $g_0^t$ since $g_k^t$ represents the true gradients of

layer $k$. By solving this problem through Lagrange dual problem (Bot et al., 2009), we derive the following outcomes,

**Theorem 3.1.** *(Divergence alleviation). If gradients are vectors, for the layers that require cross-layer gradients, their updated gradients can be expressed as,*

$$g_{opt}^t = \begin{cases} g_k^t, & \text{if } \beta \geq 0 \\ g_k^t - \theta^t g_0^t, & \text{if } \beta < 0, \end{cases} \tag{2}$$

*where $\theta^t = \frac{\beta}{\alpha}$, $\alpha = (g_0^t)^T g_0^t$ and $\beta = (g_0^t)^T g_k^t$.*

*Remark* 3.2. This theorem can be extended to the matrix form.

We provide proof for Theorem 3.1 and demonstrate how matrix gradients are incorporated into the problem in Appendix C. Our analytic solution in Equation 2 automatically determines the optimal settings for parameter $\theta^t$, eliminating the need for cumbersome manual adjustments. In our practical implementation, we consistently update the server model using the expression $g_k^t - \theta^t g_0^t$, irrespective of whether $\beta \geq 0$ or $\beta < 0$. This procedure is illustrated in Algorithm 1 in Appendix H.

*Communication Overheads.* According to the entire process, the primary process (internal cross-layer gradients) is conducted on the server. Therefore, our method does not impose any additional communication overhead between clients and the server.

## 4 CONVERGENCE ANALYSIS

In this section, we demonstrate the convergence of cross-layer gradients and propose the convergence rate in non-convex scenarios. To simplify the notations, we adopt $L_i$ to be the local objective. At first, we show the following assumptions frequently used in the convergence analysis for FL (Tan et al., 2022; Li et al., 2020b; Karimireddy et al., 2020).

**Assumption 4.1.** (Lipschitz Smooth). *Each objective function $L_i$ is L-Lipschitz smooth and satisfies that $||\nabla L_i(x) - \nabla L_i(y)|| \leq L||x - y||, \forall (x, y) \in D_i, i \in 1, ..., K$.*

**Assumption 4.2.** (Unbiased Gradient and Bounded Variance). *At each client, the stochastic gradient is an unbiased estimation of the local gradient, with $\mathbb{E}[g_i(x)] = \nabla L_i(x)$, and its variance is bounded by $\sigma^2$, meaning that $\mathbb{E}[||g_i(x) - \nabla L_i(x)||^2] \leq \sigma^2, \forall i \in 1, ..., K$, where $\sigma^2 \geq 0$.*

**Assumption 4.3.** (Bounded Expectation of Stochastic Gradients). *The expectation of the norm of the stochastic gradient at each client is bounded by $\rho$, meaning that $\mathbb{E}[||g_i(x)||] \leq \rho, \forall i \in 1, ..., K$.*

**Assumption 4.4.** (Bounded Covariance of Stochastic Gradients). *The covariance of the stochastic gradients is bounded by $\Gamma$, meaning that $Cov(g_{i,l_k}, g_{i,l_j}) \leq \Gamma, \forall i \in 1, ..., K$, where $l_k, l_j$ are the layers belonging to a model at client $i$.*

Following these assumptions, we present proof of non-convex convergence concerning the utilization of cross-layer gradients in Federated Learning (FL). We outline our principal theorems as follows.

**Theorem 4.5.** *(Per round drift). Supposed Assumption 4.1 to Assumption 4.4 are satisfied, the loss function of an arbitrary client at round $t + 1$ is bounded by,*

$$\mathbb{E}[L_{t+1,0}] \leq \mathbb{E}[L_{t,0}] - (\eta - \frac{L\eta^2}{2}) \sum_{e=0}^{E-1} ||\nabla L_{t,e}||^2 + \frac{LE\eta^2}{2}\sigma^2 + 2\eta(\Gamma + \rho^2) + L\eta^2(2\rho^2 + \sigma^2 + \Gamma). \tag{3}$$

The Theorem 4.5 demonstrates the bound of the local objective function after every communication round. Non-convex convergence can be guaranteed by the appropriate $\eta$.

**Theorem 4.6.** *(Non-convex convergence). The loss function L is monotonously decreased with the increasing communication round when,*

$$\eta < \frac{2\sum_{e=0}^{E-1}||\nabla L_{t,e}||^2 - 4(\Gamma + \rho^2)}{L(\sum_{e=0}^{E-1}||\nabla L_{t,e}||^2 + E\rho^2 + 2(2\rho^2 + \sigma^2 + \Gamma))}. \tag{4}$$

Moreover, after we prove the non-convex convergence for the cross-layer gradients, the non-convex convergence rate is described as follows.

**Theorem 4.7.** *(Non-convex convergence rate). Supposed Assumption 4.1 to Assumption 4.4 are satisfied and $\kappa = L_0 - L^*$, for an arbitrary client, given any $\epsilon > 0$, after*

$$T = \frac{2\kappa}{E\eta((2 - L\eta)\epsilon - 3L\eta\sigma^2 - 2(2 + L\eta)\Gamma - 4(1 + L\eta)\rho^2)} \tag{5}$$

Table 2: Test accuracy of model-homogeneous methods with 100 clients and sample ratio 0.1. We shade in gray the methods that are combined with our proposed method, InCo Aggregation. We bold the best results and denote the improvements compared to the original methods in red. See Appendix H.5 for the error bars of InCo methods.

| Base | Methods | Fashion-MNIST | | SVHN | | CIFAR10 | | CINIC10 | |
|---|---|---|---|---|---|---|---|---|---|
| | | $\alpha = 0.5$ | $\alpha = 1.0$ | $\alpha = 0.5$ | $\alpha = 1.0$ | $\alpha = 0.5$ | $\alpha = 1.0$ | $\alpha = 0.5$ | $\alpha = 1.0$ |
| ResNet (Stage splitting) | HeteroAvg | 87.8±1.1 | 86.0±1.0 | 85.1±2.0 | 86.9±2.3 | 64.8±2.9 | 66.7±3.3 | 48.6±2.6 | 56.5±1.6 |
| | HeteroProx | 86.8±1.5 | 83.9±1.8 | 87.8±2.1 | 89.9±1.7 | 72.5±2.1 | 73.1±1.9 | 56.4±2.0 | 60.9±1.8 |
| | HeteroScaffold | 85.2±0.8 | 86.4±0.7 | 80.6±2.3 | 86.3±2.7 | 65.5±3.0 | 69.7±2.8 | 50.8±2.9 | 57.8±3.4 |
| | HeteroNova | 84.9±1.3 | 86.7±1.1 | 84.4±1.4 | 88.0±1.7 | 60.1±3.7 | 68.0±3.5 | 46.1±2.3 | 52.1±2.2 |
| | HeteroMOON | 87.9±0.4 | 88.3±0.3 | 83.0±2.3 | 86.5±1.6 | 65.1±2.9 | 68.4±2.6 | 50.1±2.3 | 54.7± 1.8 |
| | InCoAvg | **90.2**(↑2.4) | 88.4(↑2.4) | 87.6(↑2.5) | 89.0(↑2.1) | 67.8(↑3.0) | 70.7(↑4.0) | 53.0(↑4.4) | 57.5(↑1.0) |
| | InCoProx | 88.8(↑2.0) | 86.4(↑2.5) | **89.0**(↑1.2) | **90.8**(↑0.9) | **74.5**(↑2.0) | **76.8**(↑3.7) | **59.1**(↑2.7) | **62.5**(↑1.6) |
| | InCoScaffold | 88.3(↑3.1) | **90.1**(↑3.7) | 85.4(↑4.8) | 87.8(↑1.5) | 67.3(↑1.8) | 73.8(↑4.1) | 53.5(↑2.7) | 61.7(↑3.9) |
| | InCoNova | 86.6(↑1.7) | 87.4(↑0.7) | 86.4(↑2.0) | 88.4(↑0.4) | 62.8(↑2.7) | 69.7(↑2.7) | 48.0(↑1.9) | 54.1(↑2.0) |
| | InCoMOON | 89.1(↑1.2) | 89.5(↑1.2) | 85.6(↑2.6) | 89.3(↑2.8) | 68.2(↑3.1) | 71.8(↑3.4) | 54.3(↑4.2) | 57.6(↑2.9) |
| ViT (Layer splitting) | HeteroAvg | 92.2±0.6 | 92.0±0.6 | 92.9±1.0 | 93.8±0.9 | 93.6±1.0 | 94.1±0.9 | 84.2±1.6 | 85.3±1.3 |
| | HeteroProx | 90.9±0.8 | 91.7±0.6 | 91.2±1.3 | 92.4±1.8 | 92.0±1.5 | 92.6±1.3 | 84.0±1.8 | 84.8±2.0 |
| | HeteroScaffold | 91.9±0.6 | 92.1±0.4 | 92.5±0.9 | 93.7±0.6 | 93.8±0.8 | 94.3±0.4 | 83.8±1.9 | 85.3±1.6 |
| | HeteroNova | 92.1±0.9 | 92.4±0.4 | 92.3±1.0 | 94.1±1.2 | 93.6±0.5 | 94.5±0.6 | 85.3±1.7 | 86.7±1.5 |
| | HeteroMOON | 92.0±0.4 | 92.3±0.3 | 92.7±1.1 | 94.0±0.9 | 93.5±0.8 | 94.6±0.5 | 84.7±1.4 | 85.6±1.4 |
| | InCoAvg | 93.0(↑0.8) | 93.1(↑1.1) | 94.2(↑1.3) | 95.0(↑1.2) | 94.6(↑1.0) | 95.0(↑0.9) | 85.9(↑1.7) | 86.8(↑1.5) |
| | InCoProx | 92.6(↑1.7) | 92.5(↑0.8) | 93.9(↑2.7) | 94.4(↑2.0) | 94.0(↑2.0) | 94.8(↑2.2) | 85.1 (↑1.1) | 86.0(↑1.2) |
| | InCoScaffold | 92.9(↑1.0) | 93.0(↑0.9) | 94.0(↑1.5) | 94.8(↑1.1) | 94.6(↑0.8) | 95.0(↑0.7) | 85.7(↑1.9) | 86.5(↑1.2) |
| | InCoNova | **93.1**(↑1.0) | **93.6**(↑1.2) | 94.7(↑2.4) | **95.6**(↑1.5) | **94.8**(↑1.2) | **95.7**(↑1.2) | **86.2**(↑0.9) | **88.2**(↑1.2) |
| | InCoMOON | 92.8(↑0.8) | 93.0(↑0.7) | **94.7**(↑2.0) | 95.1(↑1.1) | 94.2(↑0.7) | 95.1 (↑0.5) | 86.0(↑1.3) | 86.8(↑1.2) |

*communication rounds, we have*

$$\frac{1}{TE} \sum_{t=0}^{T-1} \sum_{e=0}^{E-1} \mathbb{E}[||\nabla L_{t,e}||^2] \le \epsilon, \; if \;\; \eta < \frac{2\epsilon - 4(\Gamma + \rho^2)}{L(\epsilon + E\rho^2 + 2(2\rho^2 + \sigma^2 + \Gamma))}. \tag{6}$$

Following these theorems, the convergence of internal cross-layer gradients is guaranteed. The proof is presented in Appendix D.

## 5 EXPERIMENTS

In this section, we conduct comprehensive experiments aimed at demonstrating three fundamental aspects: (1) the efficacy of InCo Aggregation and its extensions for various FL methods (Section 5.2), (2) the robustness analysis and ablation study of InCo Aggregation (Section 5.3), (3) in-depth analyses of the underlying principles behind InCo Aggregation (Section 5.4). Our codes are released on GitHub [3]. More experimental details and results can be found in Appendix H.

### 5.1 EXPERIMENT SETUP

*Dataset and Data Distribution.* We conduct experiments on Fashion-MNIST (Xiao et al., 2017), SVHN (Netzer et al., 2011), CIFAR-10 (Krizhevsky et al., 2009) and CINIC-10 (Darlow et al., 2018) under non-iid settings. We evaluate the algorithms under two Dirichlet distributions with $\alpha = 0.5$ and $\alpha = 1.0$ for all datasets.

*Baselines.* To demonstrate the effectiveness of InCo Aggregation, we use five baselines in model-homogeneous FL: **FedAvg** (McMahan et al., 2017), **FedProx** (Li et al., 2020b), **FedNova** (Wang et al., 2020), **Scaffold** (Karimireddy et al., 2020), and **MOON** (Li et al., 2021a) for ResNets and ViTs. In the context of model heterogeneity, we extend the training procedures of these baselines by incorporating model splitting methods, denoting the modified versions with the prefix "**Hetero**". Furthermore, by incorporating these methods with InCo Aggregation, we prefix the names with "**InCo**". Moreover, we also extend our methods to four state-of-the-art methods in model-heterogeneous FL: **HeteroFL**(Diao et al., 2021), **InclusiveFL**(Liu et al., 2022), **FedRolex**(Alam et al., 2022) and **ScaleFL**(Ilhan et al., 2023) for ResNets. We take the average accuracy of three different random seeds.

*Federated Settings.* In heterogeneous FL, we consider two architectures, ResNets and ViTs. The largest models are ResNet26 and ViT-S/12 (ViT-S with 12 layers). We deploy stage splitting for ResNets and obtain five sub-models, which can be recognized as ResNet10, ResNet14, ResNet18, ResNet22, and ResNet26. For the pre-trained ViT models, we employ layer splitting and result in five sub-models, which are ViT-S/8, ViT-S/9, ViT-S/10, ViT-S/11, and ViT-S/12. Moreover, we consider five different model capacities $\beta = \{1, 1/2, 1/4, 1/8, 1/16\}$ in hetero splitting, where for instance, $1/2$

---

[3]https://github.com/ChanYunHin/InCo-Aggregation

Table 3: Test accuracy of model-heterogeneity methods with 100 clients and sample ratio 0.1. We shade in gray the methods that are combined with our proposed method, InCo Aggregation. We denote the improvements compared to the original methods in red. See Appendix H.5 for the error bars of InCo methods.

| Base | Splitting | Methods | Fashion-MNIST | | SVHN | | CIFAR10 | | Comm. overheads | FLOPs |
|---|---|---|---|---|---|---|---|---|---|---|
| | | | $\alpha = 0.5$ | $\alpha = 1.0$ | $\alpha = 0.5$ | $\alpha = 1.0$ | $\alpha = 0.5$ | $\alpha = 1.0$ | | |
| ResNet | Hetero | HeteroFL | 88.9±1.0 | 89.7±0.7 | 90.5±1.6 | 92.2±1.3 | 65.2±3.2 | 68.4±3.6 | 4.6M | 33.4M |
| | | +InCo | 90.0(↑1.1) | 90.4(↑0.7) | 92.1(↑1.6) | 93.5(↑1.3) | 68.2(↑3.0) | 71.2(↑2.8) | 4.6M | 33.8M |
| | Stage | InclusiveFL | 89.1±1.1 | 89.8±1.0 | 88.6±2.0 | 90.0±2.2 | 65.7±3.5 | 68.4±3.3 | 12.3M | 75.2M |
| | | +InCo | 90.1(↑1.0) | 90.5(↑0.7) | 90.6(↑2.0) | 90.9(↑0.9) | 69.1(↑3.4) | 72.3(↑3.9) | 12.3M | 75.6M |
| | Hetero | FedRolex | 88.2±1.0 | 90.2±0.8 | 90.9±1.3 | 91.6±1.7 | 64.7±4.1 | 72.3±3.0 | 4.6M | 33.4M |
| | | +InCo | 90.4(↑2.2) | 91.3(↑1.1) | 92.8(↑1.9) | 93.4(↑1.8) | 67.9(↑3.2) | 75.6(↑3.3) | 4.6M | 33.8M |
| | Hetero | ScaleFL | 90.9±0.5 | 91.0±0.4 | 92.6±1.0 | 92.9±0.9 | 71.1±2.9 | 74.7±3.1 | 9.5M | 51.9M |
| | | +InCo | 91.5(↑0.6) | 91.7(↑0.7) | 93.4(↑0.8) | 93.6(↑0.7) | 73.8(↑2.7) | 76.1(↑2.4) | 9.5M | 52.3M |
| | N/A | AllSmall | 83.5±1.7 | 84.0±1.7 | 72.1±3.5 | 81.0±2.9 | 39.2±2.0 | 44.9±2.3 | 0.07M | 3.7M |
| | N/A | AllLarge | 91.8±0.5 | 92.5±0.8 | 93.4±0.8 | 93.8±0.5 | 79.6±2.9 | 82.5±1.0 | 17.5M | 112.4M |

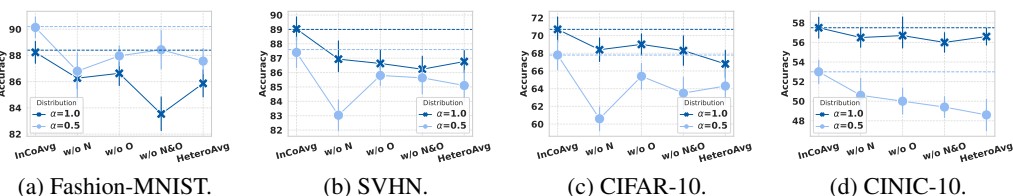

(a) Different batch sizes in CIFAR-10.  (b) Different batch sizes in CINIC-10.  (c) Different noise perturbations in CIFAR-10.  (d) Different noise perturbations in CINIC-10.

Figure 7: Robustness analysis for InCo Aggregation.

(a) Fashion-MNIST.  (b) SVHN.  (c) CIFAR-10.  (d) CINIC-10.

Figure 8: Ablation studies for InCo Aggregation. The federated settings are the same as Table 2.

indicates the widths and depths are half of the largest model ResNet26. Our experimental setup involves 100 clients, categorized into five distinct groups, with a sample ratio of 0.1. The detailed model sizes are shown in Appendix H.4.

## 5.2 INCO AGGREGATION IMPROVES ALL BASELINES.

Table 2 and Table 3 present the test accuracy of 100 clients with a sample ratio of 0.1. Table 2 provides compelling evidence for the efficacy of InCo Aggregation in enhancing the performance of all model-homogeneous baselines. Table 3 demonstrates the improvements of deploying InCo Aggregation in the model-heterogeneous methods. Moreover, Table 3 highlights that InCo Aggregation introduces no additional communication overhead and only incurs 0.4M FLOPs, which are conducted on the server side, indicating that InCo Aggregation does not impose any burden on client communication and computation resources.

## 5.3 ROBUSTNESS ANALYSIS AND ABLATION STUDY.

We delve into the robustness analysis of InCo Aggregation, examining two aspects: the impact of varying batch sizes and noise perturbations on gradients during transmission. Additionally, we perform an ablation study for InCo Aggregation. We provide more experiments in Appendix H.

*Effect of Batch Size and Noise Perturbation.* Notably, when compared to FedAvg as depicted in Figure 7a and Figure 7b, our method exhibits significant improvements while maintaining comparable performance across all settings. Furthermore, as illustrated in Figure 7c and Figure 7d, we explore the impact of noise perturbations by simulating noise with standard deviations following the gradients.

*Ablation Study.* Our ablation study includes the following methods: (i) InCoAvg w/o Normalization (HeteroAvg with cross-layer gradients and optimization), (ii) InCoAvg w/o Optimization (HeteroAvg with normalized cross-layer gradients), (iii) InCoAvg w/o Normalization and Optimization (HeteroAvg with cross-layer gradients), and (iv) HeteroAvg (FedAvg with stage splitting). The ablation study of InCo Aggregation is depicted in Figure 8, demonstrating the efficiency of InCo Aggregation.

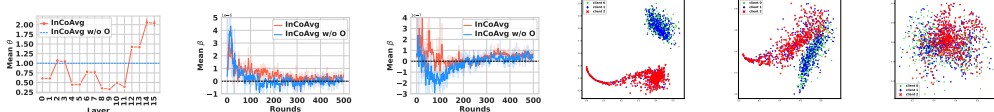

(a) $\theta$ in all layers.  (b) $\beta$ in Layer 11.  (c) $\beta$ in Layer 13.  (d) FedAvg.  (e) HeteroAvg.  (f) InCoAvg.

Figure 9: Important coefficients of Theorem 3.1 and t-SNE visualization of features. (a): $\theta$ in all layers. (b): $\beta$ in Layer 11. (c): $\beta$ in Layer 13. (d) to (f): t-SNE visualization of features learned by different methods on CIFAR-10. We select data from one class and three clients (client 0: ResNet10, client 1: ResNet14, client 2: ResNet26) to simplify the notations in t-SNE figures.

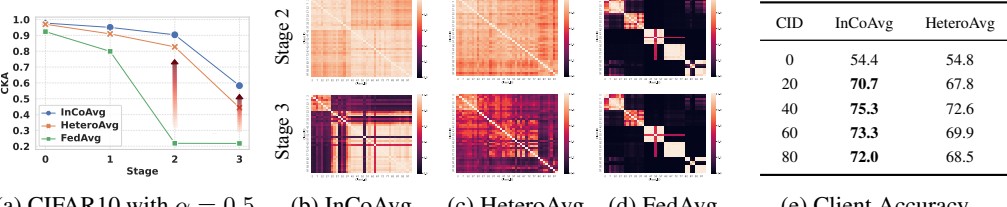

(a) CIFAR10 with $\alpha = 0.5$.  (b) InCoAvg.  (c) HeteroAvg.  (d) FedAvg.  (e) Client Accuracy.

Figure 10: CKA layer similarity, Heatmaps, and accuracy of different clients. (a): The layer similarity of different methods. (b) to (d): Heatmaps for different methods in stage 2 and stage 3. (e): Accuracy of each client group. (0: ResNet10, 20: ResNet14, 40: ResNet18, 60: ResNet22, 80: ResNet26)

## 5.4 THE REASONS FOR THE IMPROVEMENTS

We undertake a comprehensive analysis to gain deeper insights into the mechanisms underlying the efficacy of InCo Aggregation. Our analysis focuses on the following three key aspects: (1) The investigation of important coefficients $\theta$ and $\beta$ in Theorem 3.1. (2) An examination of the feature spaces generated by different methods. (3) The evaluation of CKA similarity across various layers. Moreover, we discuss the differences between adding noises and InCo gradients in Appendix H.6.

*Analysis for $\theta$ and $\beta$.* In our experiments, we set $\theta = 1$ for InCoAvg w/o Optimization, the blue dash line in Figure 9a. However, under Theorem 3.1, we observe that the value of $\theta$ varies for different layers, indicating the effectiveness of the theorem in automatically determining the appropriate $\theta$ values. $\beta > 0$ denotes the same direction between shallow layer gradients and the current layer gradients. Furthermore, Table 4 provides empirical evidence supporting the efficacy of Theorem 3.1 in heterogeneous FL.

Table 4: The Percentage of $\beta > 0$

| Methods | Percentage of $\beta > 0$ | |
|---|---|---|
| | Layer 11 | Layer 13 |
| InCoAvg | **83.8** | **74.4** |
| InCoAvg w/o O | 53.5 | 50.2 |

*t-SNE Visualizations.* Figure 9d and Figure 9e provide visual evidence of bias stemming from model heterogeneity in the FedAvg and HeteroAvg. In contrast, Figure 9f demonstrates that InCoAvg effectively addresses bias. These findings highlight the superior generalization capability of InCoAvg compared to HeteroAvg and FedAvg, indicating that InCoAvg mitigates bias issues in client models.

*Analysis for CKA Layer Similarity.* Figure 10a reveals that InCoAvg exhibits a significantly higher CKA layer similarity compared to FedAvg. Consistent with the t-SNE visualization, FedAvg's heatmaps exhibit block-wise patterns in Figure 10d due to its inability to extract features from diverse model architectures. Notably, the smallest models in InCoAvg (top left corner) exhibit lower similarity (more black) with other clients compared to HeteroAvg in stage 3. This discrepancy arises because the accuracy of the smallest models in InCoAvg is similar to that of HeteroAvg, but the performance of larger models in InCoAvg surpasses that of HeteroAvg, as indicated in Figure 10e. Consequently, a larger similarity gap emerges between the smallest models and the other models. Addressing the performance of the smallest models in InCo Aggregation represents our future research direction.

## 6 CONCLUSIONS

We propose a novel FL training scheme called InCo Aggregation, which aims to enhance the capabilities of model-homogeneous FL methods in heterogeneous FL settings. Our approach leverages normalized cross-layer gradients to promote similarity among deep layers across different clients. Additionally, we introduce a convex optimization formulation to address the challenge of gradient divergence. Through extensive experimental evaluations, we demonstrate the effectiveness of InCo Aggregation in improving heterogeneous FL performance.

ACKNOWLEDGMENTS

This work was supported by the RGC General Research Funds No. 17203320 and No. 17209822 and a seed project grant from HKU-TCL Joint Research Center for Artificial Intelligence from Hong Kong.

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

## A  CENTERED KERNEL ALIGNMENT

Centered Kernel Alignment (CKA) originally serves as a similarity measure for different kernel functions (Cortes et al., 2012). Later, its purpose has been extended to discovering meaningful similarities between internal representations of neural networks (Kornblith et al., 2019). Compared with alternative methods to monitor representation learning, such as Canonical Correlation Analysis-based methods (Raghu et al., 2017; Morcos et al., 2018) and neuron alignment methods (Li et al., 2015; Wang et al., 2018), CKA achieves the state-of-the-art performance in measuring the difference between representations of neural network. This is based on the fact that CKA reliably identifies correspondences between representations from architecturally corresponding layers in two networks trained with different initializations.[4]

Denote $X \in \mathbb{R}^{n \times p}$ and $Y \in \mathbb{R}^{n \times q}$ as two representations of $n$ data points with possibly different dimensions (i.e., $p \neq q$). These two representations fall into the following three categories: (1) internal outputs at two different layers of an individual network, (2) internal layer outputs of two architecturally identical networks trained from different initialization or by different datasets, or (3) internal layer outputs of two networks with different architectures possibly trained by different datasets. The application of CKA in our paper belongs to the third category, where we examine the CKA similarities of a corresponding layer output between every pair of local client models in the context of federated learning.

Let $k_x(\cdot, \cdot)$ and $k_y(\cdot, \cdot)$ be the kernel functions for $X$ and $Y$ respectively. Then the resulted kernel matrices of $k_x$ and $k_y$ with respect to $\mathbf{x}_1, \ldots, \mathbf{x}_n$ and $\mathbf{y}_1, \ldots, \mathbf{y}_n$ are $K_x$ and $K_y$, whose $(i, j)$-entries are $K_x(i, j) = k_x(\mathbf{x}_i, \mathbf{x}_j)$ and $K_y(i, j) = k_y(\mathbf{y}_i, \mathbf{y}_j)$. Then CKA is defined as

$$\text{CKA}(K_x, K_y) := \frac{\text{tr}(K_x H K_y H)}{\sqrt{\text{tr}(K_x H K_x H)\text{tr}(K_y H K_x H)}}, \tag{7}$$

where $H = I_n - \frac{1}{n}\mathbf{1}\mathbf{1}^{\text{T}}$ is the centering matrix.

As for the kernels in CKA, we select linear kernel (i.e., $K_x = XX^{\text{T}}$, $K_y = YY^{\text{T}}$)[5] over Radial Basis Function (RBF) kernel[6] from common kernels for the following reasons. First, experiments in Kornblith et al. (2019) manifest that linear and RBF kernels work equally well in similarity measurement of feature representations. Furthermore, it is recently validated that CKA based on an RBF kernel converges to linear CKA in the large-bandwidth limit (Alvarez, 2023). Hence, in our investigation we stick with linear CKA for computational efficiency, where the resulting linear CKA is

$$\text{CKA}_{\text{linear}}(X, Y) = \text{CKA}(XX^{\text{T}}, YY^{\text{T}})$$

$$= \frac{\text{tr}(XX^{\text{T}}HYY^{\text{T}}H)}{\sqrt{\text{tr}(XX^{\text{T}}HXX^{\text{T}}H)\text{tr}(YY^{\text{T}}HYY^{\text{T}}H)}}$$

$$= \frac{\text{tr}(Y^{\text{T}}HXX^{\text{T}}HY)}{\sqrt{\text{tr}(X^{\text{T}}HXX^{\text{T}}HX)\text{tr}(Y^{\text{T}}HYY^{\text{T}}HY)}}$$

$$= \frac{||Y^{\text{T}}HX||_{\text{F}}^2}{||X^{\text{T}}HX||_{\text{F}}||Y^{\text{T}}HY||_{\text{F}}}. \tag{8}$$

In our design, we measure the averaged CKA similarities according to the outputs from the same batch of test data. The range of CKA is between 0 and 1, and a higher CKA score means more similar paired features.

## B  COMPARISONS WITH RESIDUAL CONNECTIONS

**Remarks on self-mixture approaches in neural networks.** The goal of residual connections is to avoid exploding and vanishing gradients to facilitate the training of a single model (He et al., 2016), while cross-layer gradients aim to increase the layer similarities across a group of models that are jointly optimized in federated learning. Specifically, residual connections modify forward passes by

---

[4]We refer the readers to Section 6.1. in Kornblith et al. (2019) for a complete sanity check of representational similarity measures.

[5]Linear kernel $k(\mathbf{x}_i, \mathbf{x}_j) = \mathbf{x}_i^{\text{T}}\mathbf{x}_j$

[6]Radial Basis Function (RBF) kernel $k(\mathbf{x}_i, \mathbf{x}_j) = \exp(-\gamma||\mathbf{x}_i - \mathbf{x}_j||_2^2)$ with $\gamma > 0$

adding the shallow-layer outputs to those of the deep layers. In contrast, cross-layer gradients operate on the gradients calculated by back-propagation. We present the distinct gradient outcomes of the two methods in the following.

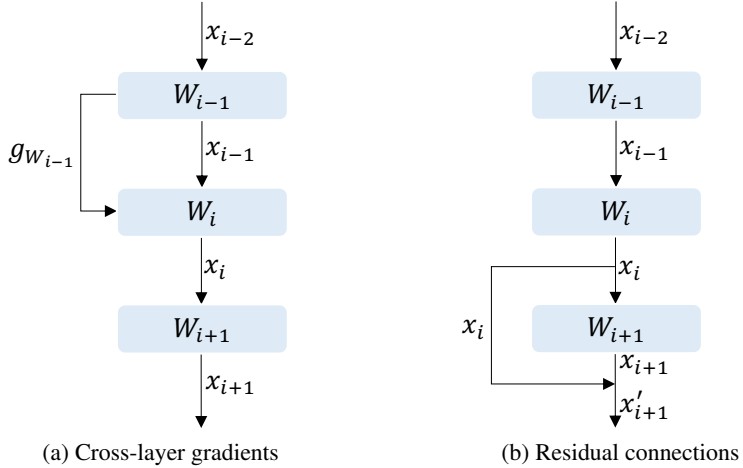

(a) Cross-layer gradients           (b) Residual connections

Figure 11: Comparison of cross-layer gradients and residual connections

Consider three consecutive layers of a feedforward neural network indexed by $i-1, i, i+1$. With a slight abuse of symbols, we use $f(\cdot; W_k)$ to denote the calculation in the $k$-th layer. Given the input $\mathbf{x_{i-2}}$ to layer $i-1$, the output from the previous layer becomes the input to the next layer, thus generating $\mathbf{x_{i-1}}, \mathbf{x_i}, \mathbf{x_{i+1}}$ sequentially.

$$
\begin{aligned}
\mathbf{x_{i-1}} &= f(\mathbf{x_{i-2}}; W_{i-1}) \\
\mathbf{x_i} &= f(\mathbf{x_{i-1}}; W_i) \\
\mathbf{x_{i+1}} &= f(\mathbf{x_i}; W_{i+1})
\end{aligned}
\tag{9}
$$

In the case of residual connections, there is an additional operation that directs $\mathbf{x_i}$ to $\mathbf{x_{i+1}}$, formulated as $\mathbf{x'_{i+1}} = \mathbf{x_{i+1}} + \mathbf{x_i}$. The gradient of $W_i$ is

$$
\begin{aligned}
g_{W_i} &= \frac{\partial loss}{\partial W_i} \\
&= \frac{\partial loss}{\partial \mathbf{x'_{i+1}}} \cdot \frac{\partial \mathbf{x'_{i+1}}}{\partial \mathbf{x_i}} \cdot \frac{\partial \mathbf{x_i}}{\partial W_i} \\
&= \frac{\partial loss}{\partial (\mathbf{x_{i+1}} + \mathbf{x_i})} \cdot \left( \frac{\partial \mathbf{x_{i+1}}}{\partial \mathbf{x_i}} + \mathbb{I} \right) \cdot \frac{\partial \mathbf{x_i}}{\partial W_i} \\
&= \frac{\partial loss}{\partial (\mathbf{x_{i+1}} + \mathbf{x_i})} \cdot \left( \frac{\partial \mathbf{x_{i+1}}}{\partial W_i} + \frac{\partial \mathbf{x_i}}{\partial W_i} \right)
\end{aligned}
\tag{10}
$$

In the case of cross-layer gradients, the gradient of $W_i$ is

$$
\begin{aligned}
g_{W_i} &= \frac{\partial loss}{\partial W_i} + \frac{\partial loss}{\partial W_{i-1}} \\
&= \frac{\partial loss}{\partial \mathbf{x}_{i+1}} \cdot \left( \frac{\partial \mathbf{x_{i+1}}}{\partial W_i} + \frac{\partial \mathbf{x_{i+1}}}{\partial W_{i-1}} \right)
\end{aligned}
\tag{11}
$$

We note that both residual connections and cross-layer gradients are subject to certain constraints. Residual connections require identical shapes for the layer outputs, while cross-layer gradients operate on the layer weights with the same shape.

## C   PROOF OF THEOREM 3.1

This section demonstrates the details of the proof of Theorem 3.1. We will present the proof of Theorem 3.1 in the vector form and the matrix form.

## C.1  VECTOR FORM

We state the convex optimization problem Theorem 1 in the vector form in the following,

$$\min_{g_{opt}} \quad ||g_k - g_{opt}||_2^2,$$
$$s.t. \quad \langle g_{opt}, g_0 \rangle \geq 0. \tag{12}$$

Because the superscript $t$ would not influence the proof of the theorem, we simplify the notation $g^t$ to $g$. We use Equation 12 instead of Equation 1 to complete this proof. The Lagrangian of Equation 12 is shown as,

$$
\begin{aligned}
L(g_{opt}, \lambda) =& (g_k - g_{opt})^T (g_k - g_{opt}) - \lambda g_{opt}^T g_0 \\
=& g_k^T g_k - g_{opt}^T g_k - g_k^T g_{opt} + g_{opt}^T g_{opt} - \lambda g_{opt}^T g_0 \\
=& g_k^T g_k - 2 g_{opt}^T g_k + g_{opt}^T g_{opt} - \lambda g_{opt}^T g_0.
\end{aligned}
\tag{13}
$$

Let $\frac{\partial L(g_{opt}, \lambda)}{\partial g_{opt}} = 0$, we have

$$g_{opt} = g_k + \lambda g_0 / 2, \tag{14}$$

which is the optimum point for the primal problem Equation 12. To get the Lagrange dual function $L(\lambda) = \inf_{g_{opt}} L(g_{opt}, \lambda)$, we substitute $g_{opt}$ by $g_k + \lambda g_0 / 2$ in $L(g_{opt}, \lambda)$. We have

$$
\begin{aligned}
L(\lambda) =& g_k^T g_k - 2(g_k + \frac{\lambda g_0}{2})^T g_k + (g_k + \frac{\lambda g_0}{2})^T (g_k + \frac{\lambda g_0}{2}) - \lambda (g_k + \frac{\lambda g_0}{2})^T g_0 \\
=& g_k^T g_k - 2 g_k^T g_k - \lambda g_0^T g_k + g_k^T g_k + \frac{\lambda g_0^T g_k}{2} + \frac{\lambda g_k^T g_0}{2} + \frac{\lambda^2 g_0^T g_0}{4} - \lambda g_k^T g_0 - \frac{\lambda^2 g_0^T g_0}{2} \\
=& g_k^T g_k - 2 g_k^T g_k + g_k^T g_k - \lambda g_0^T g_k + \lambda g_0^T g_k + \frac{\lambda^2 g_0^T g_0}{4} - \frac{\lambda^2 g_0^T g_0}{2} - \lambda g_k^T g_0 \\
=& -\frac{g_0^T g_0}{4} \lambda^2 - g_k^T g_0 \lambda.
\end{aligned}
\tag{15}
$$

Thus, the Lagrange dual problem is described as follows,

$$\max_\lambda L(\lambda) = -\frac{g_0^T g_0}{4} \lambda^2 - g_k^T g_0 \lambda,$$
$$s.t. \ \lambda \geq 0. \tag{16}$$

$L(\lambda)$ is a quadratic function. Because $g_0^T g_0 \geq 0$, the maximum of $L(\lambda)$ is at the point $\lambda = -\frac{2b}{a}$ where $a = g_0^T g_0$ and $b = g_k^T g_0$ if we do not consider the constraint. It is clear that this convex optimization problem holds strong duality because it satisfies Slater's constraint qualification(Boyd et al., 2004), which indicates that the optimum point of the dual problem Equation 16 is also the optimum point for the primal problem Equation 12. We substitute $\lambda$ by $-\frac{2b}{a}$ in Equation 14, and we have

$$
g_{opt} = \begin{cases} g_k, & \text{if } b \geq 0, \\ g_k - \theta^t g_0, & \text{if } b < 0, \end{cases}
\tag{17}
$$

where $\theta^t = \frac{b}{a}$, $a = (g_0)^T g_0$ and $b = g_k^T g_0$. We add the superscript $t$ to all gradients, and we finish the proof of Theorem 3.1.

## C.2  MATRIX FORM

The proof of the matrix form is similar to Appendix C.1. We update Equation 12 to the matrix form as follows,

$$\min_{G_{opt}} \quad ||G_k - G_{opt}||_F^2,$$
$$s.t. \quad \langle G_{opt}, G_0 \rangle \geq 0. \tag{18}$$

Similar to Equation 13, the Lagragian of Equation 18 is,

$$L(G_{opt}, \lambda) = tr(G_k^T G_k) - tr(G_{opt}^T G_k) - tr(G_k^T G_{opt}) + tr(G_{opt}^T G_{opt}) - \lambda tr(G_{opt}^T G_0), \tag{19}$$

where $tr(A)$ means the trace of the matrix $A$. We can obtain the optimum point for Equation 18 according to $\frac{\partial L(G_{opt}, \lambda)}{\partial G_{opt}} = 0$. We have

$$G_{opt} = G_k + \lambda G_0/2. \tag{20}$$

Similar to the analysis in Appendix C.1 and Equation 15, we get the Lagrange dual problem as follows,

$$\max_{\lambda} L(\lambda) = -\frac{tr(G_0^T G_0)}{4} \lambda^2 - tr(G_k^T G_0)\lambda, \tag{21}$$
$$s.t. \ \lambda \geq 0,$$

where $tr(G_0^T G_0) \geq 0$. Following the same analysis in Appendix C.1, we have

$$G_{opt} = \begin{cases} G_k, & \text{if } b \geq 0, \\ G_k - \theta^t G_0, & \text{if } b < 0, \end{cases} \tag{22}$$

where $\theta^t = \frac{b}{a}$, $a = tr(G_0^T G_0)$ and $b = tr(G_k^T G_0)$. At last, we have finished the proof of the matrix form of Theorem 3.1.

## D   PROOF OF CONVERGENCE ANALYSIS

We show the details of convergence analysis for cross-layer gradients. $W_{t,e}^{l_i}$ are the weights from the layers which need cross-layer gradients at round $t$ of the local step $e$. To simplify the notations, we use $W_{t,e}$ instead of $W_{t,e}^{l_i}$.

**Lemma D.1.** *(Per Round Progress.) Suppose our functions satisfy Assumption 4.1 and Assumption 4.2. The expectation of a loss function of any arbitrary clients at communication round $t$ after $E$ local steps are bounded as,*

$$\mathbb{E}[L_{t,E-1}] \leq \mathbb{E}[L_{t,0}] - (\eta - \frac{L\eta^2}{2}) \sum_{e=0}^{E-1} ||\nabla L_{t,e}||^2 + \frac{LE\eta^2}{2} \sigma^2. \tag{23}$$

*Proof.*

Considering an arbitrary client, we omit the client index $i$ in this lemma. Let $W_{t,e+1} = W_{t,e} - \eta g_{t,e}$, we have

$$L_{t,e+1} \leq L_{t,e} + \langle \nabla L_{t,e}, W_{t,e+1} - W_{t,e} \rangle + \frac{L}{2} ||W_{t,e+1} - W_{t,e}||^2$$
$$\leq L_{t,e} - \eta \langle \nabla L_{t,e}, g_{t,e} \rangle + \frac{L}{2} ||\eta g_{t,e}||^2, \tag{24}$$

where Equation 24 follows Assumption 4.1. We take expectation on both sides of Equation 24, then

$$\mathbb{E}[L_{t,e+1}] \leq \mathbb{E}[L_{t,e}] - \eta \mathbb{E}[\langle \nabla L_{t,e}, g_{t,e} \rangle] + \frac{L}{2} \mathbb{E}[||\eta g_{t,e}||^2]$$
$$= \mathbb{E}[L_{t,e}] - \eta ||\nabla L_{t,e}||^2 + \frac{L\eta^2}{2} \mathbb{E}[||g_{t,e}||^2]$$
$$\stackrel{(a)}{=} \mathbb{E}[L_{t,e}] - \eta ||\nabla L_{t,e}||^2 + \frac{L\eta^2}{2} (\mathbb{E}[||g_{t,e}||]^2 + Var(||g_{t,e}||))$$
$$= \mathbb{E}[L_{t,e}] - \eta ||\nabla L_{t,e}||^2 + \frac{L\eta^2}{2} (||\nabla L_{t,e}||^2 + Var(||g_{t,e}||))$$
$$\stackrel{(b)}{\leq} \mathbb{E}[L_{t,e}] - (\eta - \frac{L\eta^2}{2}) ||\nabla L_{t,e}||^2 + \frac{L\eta^2}{2} \sigma^2, \tag{25}$$

where (a) follows $Var(X) = \mathbb{E}[X^2] - \mathbb{E}^2[X]$, and (b) is Assumption 4.2. Telescoping local step 0 to $E-1$, we have

$$\mathbb{E}[L_{t,E-1}] \leq \mathbb{E}[L_{t,0}] - (\eta - \frac{L\eta^2}{2}) \sum_{e=0}^{E-1} ||\nabla L_{t,e}||^2 + \frac{LE\eta^2}{2} \sigma^2, \tag{26}$$

then we finish the proof of Lemma D.1.

**Lemma D.2.** *(Bound Client Dirft.) Suppose our functions satisfy Assumption 4.2, Assumption 4.3 and Assumption 4.4. After each aggregation, the updates, $\Delta W$, for the layers need cross-layer gradients have bounded drift:*

$$\mathbb{E}[||\Delta W||^2] \leq 2\eta^2(2\rho^2 + \sigma^2 + \Gamma). \tag{27}$$

*Proof.*

We have $W_{t+1,0} - W_{t,E-1} = \Delta W = \eta(g_{l_0} + g_{l_i}), \forall l_i$ need cross-layer gradients. Because all gradients are in the same aggregation round, we omit the time subscript in this proof process. Since $\eta$ is a constant, we also simplify it. $g_{l_0}$ and $g_{l_i}$ the gradients from the same client, indicating that they are dependent, then

$$||\Delta W||^2 = ||g_{l_0} + g_{l_i}||^2$$
$$\overset{(c)}{\leq} ||g_{l_0}||^2 + 2||\langle g_{l_0}, g_{l_i}\rangle|| + ||g_{l_i}||^2, \tag{28}$$

where (c) is Cauchy–Schwarz inequality. We take the expectation on both sides, then

$$\mathbb{E}[||\Delta W||^2] \leq \mathbb{E}[||g_{l_0}||^2] + 2\mathbb{E}[||\langle g_{l_0}, g_{l_i}\rangle||] + \mathbb{E}[||g_{l_i}||^2]$$
$$\overset{(a)}{=} \mathbb{E}[||g_{l_0}||]^2 + Var(||g_{l_0}||) + \mathbb{E}[||g_{l_i}||]^2 + Var(||g_{l_i}||) + 2\mathbb{E}[||\langle g_{l_0}, g_{l_i}\rangle||]$$
$$\overset{(d)}{\leq} 2(\rho^2 + \sigma^2) + 2\mathbb{E}[||g_{l_0}, g_{l_i}||] \tag{29}$$
$$\overset{(e)}{=} 2(\rho^2 + \sigma^2) + 2(Cov(g_{l_0}, g_{l_i}) + \mathbb{E}[||g_{l_0}||]\mathbb{E}[||g_{l_i}||])$$
$$\overset{(f)}{\leq} 2(\rho^2 + \sigma^2) + 2(\Gamma + \rho^2)$$
$$= 4\rho^2 + 2\sigma^2 + 2\Gamma,$$

where (d) follows assumption Assumption 4.2 and Assumption 4.3, (e) follows the covariance formula, and (f) follows assumption Assumption 4.4. We put back $\eta^2$ to the final step of Equation 29. At last, we complete the proof of Lemma D.2.

### D.1 PROOF OF THEOREM 4.5 AND THEOREM 4.6

We state Theorem 4.5 again in the following,

*(Per round drift) Supposed Assumption 4.1 to Assumption 4.4 are satisfied, the loss function of an arbitrary client at round $t + 1$ is bounded by,*

$$\mathbb{E}[L_{t+1,0}] \leq \mathbb{E}[L_{t,0}] - (\eta - \frac{L\eta^2}{2}) \sum_{e=0}^{E-1} ||\nabla L_{t,e}||^2 +$$
$$\frac{LE\eta^2}{2}\sigma^2 + 2\eta(\Gamma + \rho^2) + L\eta^2(2\rho^2 + \sigma^2 + \Gamma). \tag{30}$$

*Proof.*

Following the Assumption 4.1, we have

$$L_{t+1,0} \leq L_{t,E-1} + \langle \nabla L_{t,E-1}, W_{t+1,0} - W_{t,E-1}\rangle + \frac{L}{2}||W_{t+1,0} - W_{t,E-1}||^2$$
$$= L_{t,E-1} + \eta\langle \nabla L_{t,E-1}, g_{l_0} + g_{l_1}\rangle + \frac{L}{2}\eta^2||\Delta W||^2. \tag{31}$$

Taking the expectation on both sides, we obtain

$$\mathbb{E}[L_{t+1,0}] = \mathbb{E}[L_{t,E-1}] + \eta\mathbb{E}[\langle \nabla L_{t,E-1}, g_{l_0} + g_{l_1}\rangle] + \frac{L}{2}\eta^2\mathbb{E}[||\Delta W||^2]. \tag{32}$$

The first item is Lemma D.1, and the third item is Lemma D.2. We consider the second item $\mathbb{E}[\langle \nabla L_{t,E-1}, g_{l_0} + g_{l_1}\rangle]$ in the following, then,

$$\mathbb{E}[\langle \nabla L_{t,E-1}, g_{l_0} + g_{l_1}\rangle] = \mathbb{E}[\nabla L_{t,E-1}g_{l_0}] + \mathbb{E}[\nabla L_{t,E-1}g_{l_1}]$$
$$\overset{(e)}{=} Cov(\nabla L_{t,E-1}, g_{l_0}) + \mathbb{E}[||\nabla L_{t,E-1}||]\mathbb{E}[||g_{l_0}||] +$$
$$Cov(\nabla L_{t,E-1}, g_{l_1}) + \mathbb{E}[||\nabla L_{t,E-1}||]\mathbb{E}[||g_{l_1}||] \tag{33}$$
$$\overset{(f)}{\leq} 2(\Gamma + \rho^2).$$

Combining two lemmas and Equation 33, we have

$$\mathbb{E}[L_{t+1,0}] \leq \mathbb{E}[L_{t,0}] - (\eta - \frac{L\eta^2}{2}) \sum_{e=0}^{E-1} ||\nabla L_{t,e}||^2 + \frac{LE\eta^2}{2}\sigma^2 + 2\eta(\Gamma + \rho^2) + L\eta^2(2\rho^2 + \sigma^2 + \Gamma),$$ (34)

then we finish the proof of Theorem 4.5.

For Theorem 4.6, we consider the sum of the second term to the last term in Equation 34 to be smaller than 0, i.e.,

$$-(\eta - \frac{L\eta^2}{2}) \sum_{e=0}^{E-1} ||\nabla L_{t,e}||^2 + \frac{LE\eta^2}{2}\sigma^2 + 2\eta(\Gamma + \rho^2) + L\eta^2(2\rho^2 + \sigma^2 + \Gamma) < 0, \quad (35)$$

then, we have

$$\eta < \frac{2\sum_{e=0}^{E-1} ||\nabla L_{t,e}||^2 - 4(\Gamma + \rho^2)}{L(\sum_{e=0}^{E-1} ||\nabla L_{t,e}||^2 + E\rho^2 + 2(2\rho^2 + \sigma^2 + \Gamma))}. \quad (36)$$

We finish the proof of Theorem 4.6.

### D.2 PROOF OF THEOREM 4.7

Telescoping the communication rounds from $t = 0$ to $t = T - 1$ with the local step from $e = 0$ to $e = E - 1$ on the expectation on both sides of Equation 34, we have

$$\frac{1}{TE} \sum_{t=0}^{T-1} \sum_{e=0}^{E-1} ||\nabla L_{t,e}||^2 \leq \frac{\frac{2}{TE}\sum_{t=0}^{T-1}(\mathbb{E}[L_{t,0}] - \mathbb{E}[L_{t+1,0}]) + L\eta^2\sigma^2 + 4\eta(\Gamma + \rho^2) + 2L\eta^2(2\rho^2 + \sigma^2 + \Gamma)}{2\eta - L\eta^2}.$$ (37)

Given any $\epsilon > 0$, let

$$\frac{\frac{2}{TE}\sum_{t=0}^{T-1}(\mathbb{E}[L_{t,0}] - \mathbb{E}[L_{t+1,0}]) + L\eta^2\sigma^2 + 4\eta(\Gamma + \rho^2) + 2L\eta^2(2\rho^2 + \sigma^2 + \Gamma)}{2\eta - L\eta^2} \leq \epsilon, \quad (38)$$

and we denote $\kappa = L_0 - L^*$, then Equation 38 becomes

$$\frac{\frac{2\kappa}{TE} + L\eta^2\sigma^2 + 4\eta(\Gamma + \rho^2) + 2L\eta^2(2\rho^2 + \sigma^2 + \Gamma)}{2\eta - L\eta^2} \leq \epsilon, \quad (39)$$

because $\sum_{t=0}^{T-1}(\mathbb{E}[L_{t,0}] - \mathbb{E}[L_{t+1,0}]) \leq \kappa$. We consider $T$ in Equation 39, i.e.,

$$T \geq \frac{2\kappa}{E\eta((2 - L\eta)\epsilon - 3L\eta\sigma^2 - 2(2 + L\eta)\Gamma - 4(1 + L\eta)\rho^2)}, \quad (40)$$

then, we have

$$\frac{1}{TE} \sum_{t=0}^{T-1} \sum_{e=0}^{E-1} ||\nabla L_{t,e}||^2 \leq \epsilon, \quad (41)$$

when

$$\eta < \frac{2\epsilon - 4(\Gamma + \rho^2)}{L(\epsilon + E\rho^2 + 2(2\rho^2 + \sigma^2 + \Gamma))}. \quad (42)$$

We complete the proof of Theorem 4.7.

## E  MORE RELATED WORKS

### E.1  FEDERATED LEARNING.

In 2017, Google proposed a novel machine learning technique, i.e., Federated Learning (FL), to organize collaborative computing among edge devices or servers (McMahan et al., 2017). It enables multiple clients to collaboratively train models while keeping training data locally, facilitating privacy protection. Various synchronous or asynchronous FL schemes have been proposed and achieved good performance in different scenarios. For example, FedAvg (McMahan et al., 2017) takes a weighted average of the models trained by local clients and updates the local models iteratively. FedProx (Li et al., 2020b) generalized and re-parametrized FedAvg, guaranteeing the convergence when learning over non-iid data. FedAsyn (Xie et al., 2020) employed coordinators and schedulers to achieve an asynchronous training process.

### E.2 HETEROGENEOUS MODELS.

The clients in homogeneous federated learning frameworks have identical neural network architectures, while the edge devices or servers in real-world settings show great diversity. They usually have different memory and computation capabilities, making it difficult to deploy the same machine-learning model in all the clients. Therefore, researchers have proposed various methods supporting heterogeneous models in the FL environment.

**Knowledge Distillation.** Knowledge distillation (KD) (Hinton et al., 2015) was proposed by Hinton et al., aiming to train a student model with the knowledge distilled from a teacher model, which becomes an important research area in Machine Learning (Gou et al., 2021; Zhao et al., 2023). Inspired by the knowledge distillation, several studies(Li & Wang, 2019; Li et al., 2021b; He et al., 2020) are proposed to address the system heterogeneity problem. In FedMD(Li & Wang, 2019), the clients distill and transmit logits from a large public dataset, which helps them learn from both logits and private local datasets. In RHFL (Fang & Ye, 2022), the knowledge is distilled from the unlabeled dataset and the weights of clients are computed by the symmetric cross-entropy loss function. Unlike the aforementioned methods, data-free KD is a new approach to completing the knowledge distillation process without the training data. The basic idea is to optimize noise inputs to minimize the distance to prior knowledge(Nayak et al., 2019). Chen et al.(Chen et al., 2019) train Generative Adversarial Networks (GANs)(Goodfellow et al., 2014) to generate training data for the entire KD process utilizing the knowledge distilled from the teacher model. In FedHe(Chan & Ngai, 2021), a server directly averages the logits transmitted from clients. FedGen(Zhu et al., 2021) adopts a generator to simulate the prior knowledge from all the clients, which is used along with the private data from clients in local training. In FedGKT(He et al., 2020), a neural network is separated into two segments, one held by clients, the other preserved in a server, in which the features and logits from clients are sent to the server to train the large model. In Felo (Chan & Ngai, 2022), the representations from the intermediate layers are the knowledge instead of directly using the logits.

**Public or Generated Data.** In FedML (Shen et al., 2020), latent information from homogeneous models is applied to train heterogeneous models. FedAUX (Sattler et al., 2021) initialized heterogeneous models by unsupervised pre-training and unlabeled auxiliary data. FCCL (Huang et al., 2022) calculate a cross-correlation matrix according to the global unlabeled dataset to exchange knowledge. However, these methods require a public dataset. The server might not be able to collect sufficient data due to data availability and privacy concerns.

**Model Compression.** Although HeteroFL (Diao et al., 2021) derives local models with different sizes from one large model, the architectures of local and global models still have to share the same model architecture, and it is inflexible that all models have to be retrained when the best participant joins or leaves the FL training process. Federated Dropout (Caldas et al., 2018) randomly selects sub-models from the global models following the dropout way. SlimFL (Baek et al., 2022) incorporated width-adjustable slimmable neural network (SNN) architectures(Yu & Huang, 2019) into FL which can tune the widths of local neural networks. FjORD (Horvath et al., 2021) tailored model widths to clients' capabilities by leveraging Ordered Dropout and a self-distillation methodology. FedRoLex (Alam et al., 2022) proposes a rolling sub-model extraction scheme to adapt to the heterogeneous model environment. However, similar to HeteroFL, they only vary the number of parameters for each layer.

## F PROBLEM FORMULATION

In this section, we introduce federated learning with model heterogeneity. Federated learning aims to foster collaboration with clients to jointly train a shared global model while preserving the privacy of their local data. However, in the context of model heterogeneity, it becomes challenging to maintain the same architectures across all clients. Specifically, we consider a set of physical resources denoted as $\{R_i\}_{i=1}^n$, where $R_i$ represents the available resources of client $i$. For each local client model $\{M_i\}_{i=1}^n$, the resource requirement $R(M_i)$ must be smaller than or equal to the available resources of client $i$, i.e. $\{R(M_i) \leq R_i\}_{i=1}^n$. To satisfy this constraint, the client models have varying sizes and architectures. Let $w_i$ denote the weights of the client model $M_i$, and $f(x, w_i)$ represent the forward function of model $M_i$ with input $x$. Moreover, each client has a local dataset $D_i = \{(x_{k,i}, y_{k,i}) | k \in \{1, 2, ..., |D_i|\}\}$, where $|D_i|$ signifies the size of a dataset $D_i$. The loss

function $l_i$ of client $i$ is shown as follows,

$$\min_{w} \quad l_i(w_i) = \frac{1}{|D_i|} \sum_{k=1}^{|D_i|} l_{CSE}(f(x_k; w_i), y_k), \tag{43}$$

where $l_{CSE}$ is a cross-entropy function. Moreover, if we denote $K = \sum_{i=1}^{n} |D_i|$ as the total size of all local datasets, the global optimization problem is,

$$\min_{w_1, w_2, ..., w_n} L(w_1, ..., w_n) = \sum_{i=1}^{n} \frac{|D_i|}{K} l_i(w_i), \tag{44}$$

where the optimized model weights $\{w_1, w_2, ..., w_n\}$ are the parameters from $M_{i_{i=1}}^{n}$. In our case, $\{w_1, w_2, ..., w_n\}$ are split from the server model weight $w_s$ from the server model $M_s$. The goal of our paper is to propose a method that can effectively optimize Equation 44.

## G  CONFIGURATIONS AND MORE RESULTS OF THE CASE STUDY

### G.1  CONFIGURATIONS

In the case study, we have five ResNet models which are stage splitting from ResNet26, resulting in ResNet10, ResNet14, ResNet18, ResNet22, and ResNet26. Five ViTs models are ViT-S/8, ViT-S/9, ViT-S/10, ViT-S/11, ViT-S/12, the results from the layer splitting of ViT-S/12. The model prototypes are the same as the experiment settings. To quantify a model's degree of bias towards its local dataset, we use CKA similarities among the clients based on the outputs from the same stages in ResNet (ResNets of different sizes always contain four stages) and the outputs from the same layers in Vision Transformers (ViTs) (Dosovitskiy et al., 2020). Specifically, we measure the averaged CKA similarities according to the outputs from the same batch of test data. The range of CKA is between 0 and 1, and a higher CKA score means more similar paired features. We train FedAvg under three settings: IID with the homogeneous setting, Non-IID with the homogeneous setting, and Non-IID with the heterogeneous setting. FedAvg only aggregates gradients from the models sharing the same architectures under the heterogeneous model setting (Lin et al., 2020). For ResNets, we conduct training 100 communication rounds, while only 20 rounds for ViTs. The local training epochs for clients are five for all settings. We use Adam(Kingma & Ba, 2015) optimizer with default parameter settings for all client models, and the batch size is 64. We use two small federated scales. One is ten clients deployed the same model architecture (ResNet18 for ResNets and ViT-S/12 for ViTs), which is called a homogeneous setting. The other is ten clients with five different model architectures, which is a heterogeneous setting. This setting means that we have five groups whose architectures are heterogeneous, but the clients belonging to the same group have the same architectures.

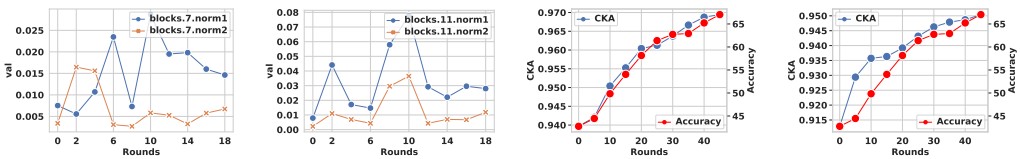

(a) Similarity of gradients in Block 7.  (b) Similarity of gradients in Block 11.  (c) The relations between CKA and accuracy in stage1.  (d) The relations between CKA and accuracy in stage2.

Figure 12: Cross-environment similarity and more results between accuracy and CKA similarity. (a) and (b): Cross-environment similarity from Block 7 and Block 11 from ViTs. (c) and (d): The positive relation between stage1 and stage2.

### G.2  RELATIONS BETWEEN CKA AND ACCURACY.

In this subsection, we continue to describe more results about the relations between CKA similarity and accuracy. Similar to Figure 1c from stage0, Figure 12c and Figure 12d show the positive relations between CKA and accuracy from stage1 and stage2.

### G.3  GRADIENT ANALYSIS FOR VITS

Similar to the gradient analyses conducted for ResNets, we have performed the analysis of gradient distributions for ViTs. In our investigation, we have analyzed the outputs from the norm1 and norm2 layers within the ViT blocks and have also applied InCo Aggregation to these layers. The selection

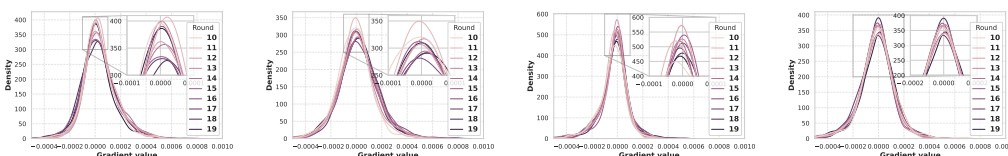

(a) Block7 norm1 in IID with homo.
(b) Block7 norm2 in IID with homo.
(c) Block11 norm1 in IID with homo.
(d) Block11 norm2 in IID with homo.

Figure 13: The gradient distributions from round 10 to 20 of ViTs in IID with homo.

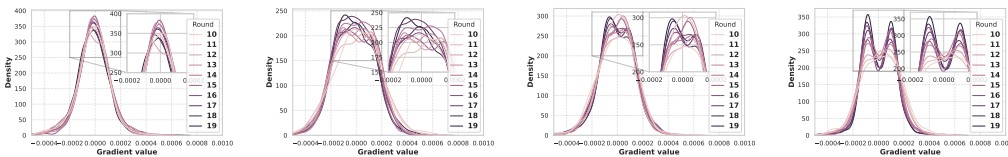

(a) Block7 norm1 in Non-IID with hetero.
(b) Block7 norm2 in Non-IID with hetero.
(c) Block11 norm1 in Non-IID with hetero.
(d) Block11 norm2 in Non-IID with hetero.

Figure 14: The gradient distributions from round 10 to 20 of ViTs in Non-IID with hetero.

of norm1 and norm2 layers is motivated by the significance of Layer Norm in the architecture of transformers (Xiong et al., 2020). Additionally, we have chosen Block7 and Block11 for analysis as, in the context of heterogeneous models, Block7 is the final layer of the smallest ViTs, while Block11 represents the final layer of the largest ViTs.

From Figure 12a and Figure 12b, we observe that the cross-environment similarities derived from the shallow layer norm (norm1) are higher compared to those from the deep layer norm (norm2). Moreover, similar to the analysis conducted for ResNets, we discover that the distributions of norm1 in ViTs exhibit greater smoothness compared to norm2, as depicted in Figure 13 and Figure 14. These findings reinforce the notion that InCo Aggregation is indeed suitable for ViTs.

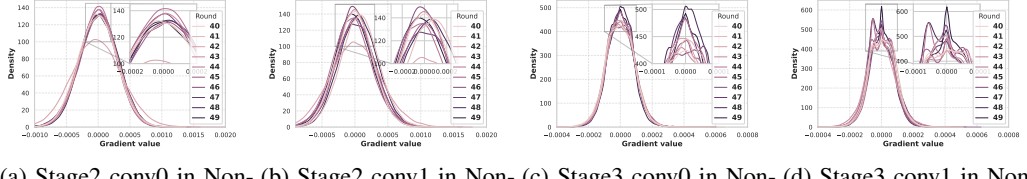

(a) Stage2 conv0 in Non-IID with hetero.
(b) Stage2 conv1 in Non-IID with hetero.
(c) Stage3 conv0 in Non-IID with hetero with different seed.
(d) Stage3 conv1 in Non-IID with hetero with different seed.

Figure 15: The gradient distributions from round 40 to 50 of ResNets in Non-IID with hetero. (a) and (b) Stage2 conv0 and conv1. (c) and (d) Stage3 conv0 and conv1 with different seed.

### G.4 Gradient Distributions from Stage 2 and Different Seed.

Figure 15a and Figure 15b demonstrate the gradient distributions in Stage 2. In contrast to the gradient distributions of Stage 3, the differences in gradient distributions across different layers are less evident for Stage 2. This can be observed from Figure 1a, where the CKA similarity for Stage 2 is considerably higher than that of Stage 3. The higher similarity indicates that Stage 2 is relatively less biased and more generalized compared to Stage 3, resulting in less noticeable differences in gradient distributions. This observation further supports the relationship between similarity and smoothness, as higher similarity leads to smoother distributions. Moreover, Figure 15c and Figure 15d illustrate that the gradient distributions still keep the same properties in different random seed, indicating that the relations between similarity and smooth gradients are not affected by SGD noise.

### G.5 Other Relations between CKA and the Statistics of Gradients.

Figure 16 provides an overview of additional relationships between layer similarity and the cross-environment gradient statistics derived from IID with homo and Non-IID with hetero. We calculate the difference between gradients from the same layer across these two environments. To clarify the tendency of similarity for each stage, we normalize the results according to the smallest value within each stage. As shown in Figure 16, none of these gradient statistics exhibit stronger correlations with the similarity of gradients compared to the smoothness, discussed in Section 2.2 and Appendix G.3.

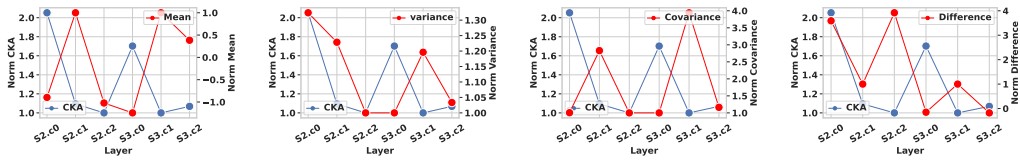

(a) CKA with mean.     (b) CKA with variance.     (c) CKA with covariance.     (d) CKA with difference.

Figure 16: Other relationships between CKA and the cross-environment statistics of gradients from IID with homo and Non-IID with hetero. We use abbreviations for "stage" and "conv," represented as "s" and "c" respectively. For example, "s2c0" represents stage 2, conv0.

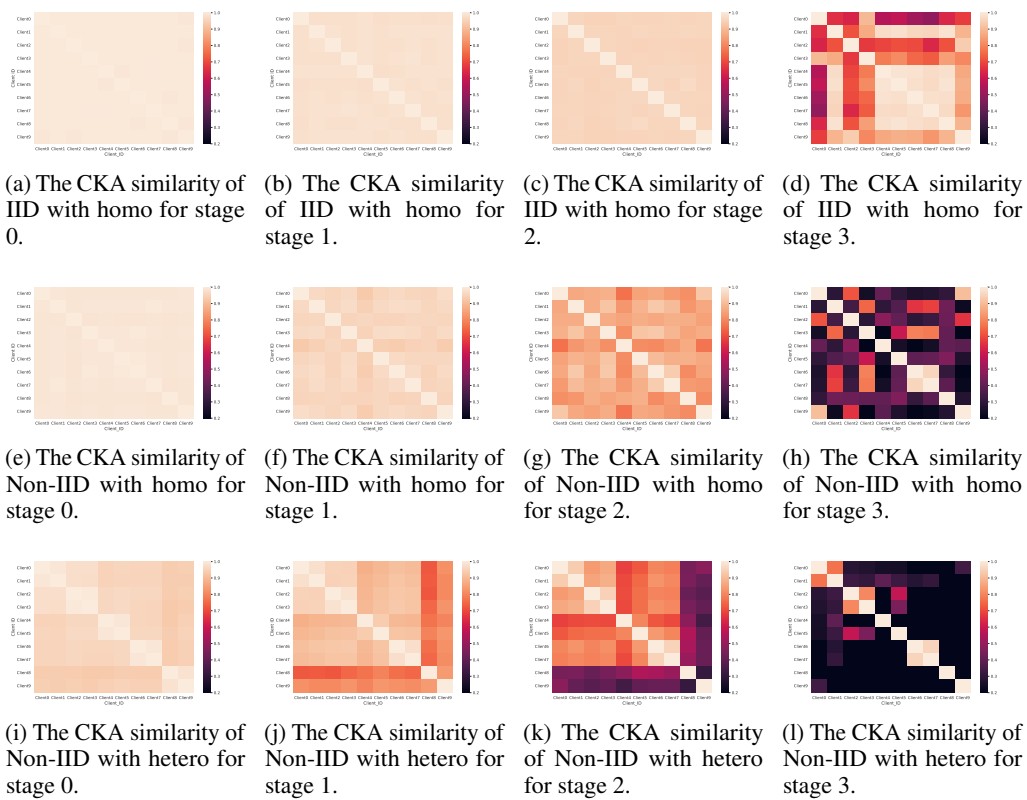

(a) The CKA similarity of IID with homo for stage 0.

(b) The CKA similarity of IID with homo for stage 1.

(c) The CKA similarity of IID with homo for stage 2.

(d) The CKA similarity of IID with homo for stage 3.

(e) The CKA similarity of Non-IID with homo for stage 0.

(f) The CKA similarity of Non-IID with homo for stage 1.

(g) The CKA similarity of Non-IID with homo for stage 2.

(h) The CKA similarity of Non-IID with homo for stage 3.

(i) The CKA similarity of Non-IID with hetero for stage 0.

(j) The CKA similarity of Non-IID with hetero for stage 1.

(k) The CKA similarity of Non-IID with hetero for stage 2.

(l) The CKA similarity of Non-IID with hetero for stage 3.

Figure 17: The CKA similarity of IID with homo, Non-IID with homo and Non-IID with hetero for ResNets.

### G.6 HEATMAPS FOR THE CASE STUDY

In this part, we will show the heatmaps for all stages of ResNets and layer 4 to layer 7 of ViTs in Figure 17 and Figure 18. These heatmaps are the concrete images for Figure 1. We can see that the CKA similarity is lower with the deeper stages or layers no matter in ResNets and ViTs. However, it is notable that the different patterns for CKA similarity between ResNets and ViTs from the comparison between Figure 17 and Figure 18. To get a clear analysis, we focus on the last stage of ResNets and layer 7 of ViTs, which are the most biased part of the entire model. Like Figure 17l in ResNets, almost all clients are dissimilar, while only a part of clients has low similarity in ViTs from Figure 18l. Along with the experiment results from Table 2, the improvements in ViTs from FedInCo are modest. One possible reason is that we neglect more biased clients and regard all clients as having the same level of bias in ViTs, which is a possible improvement for FedInCo.

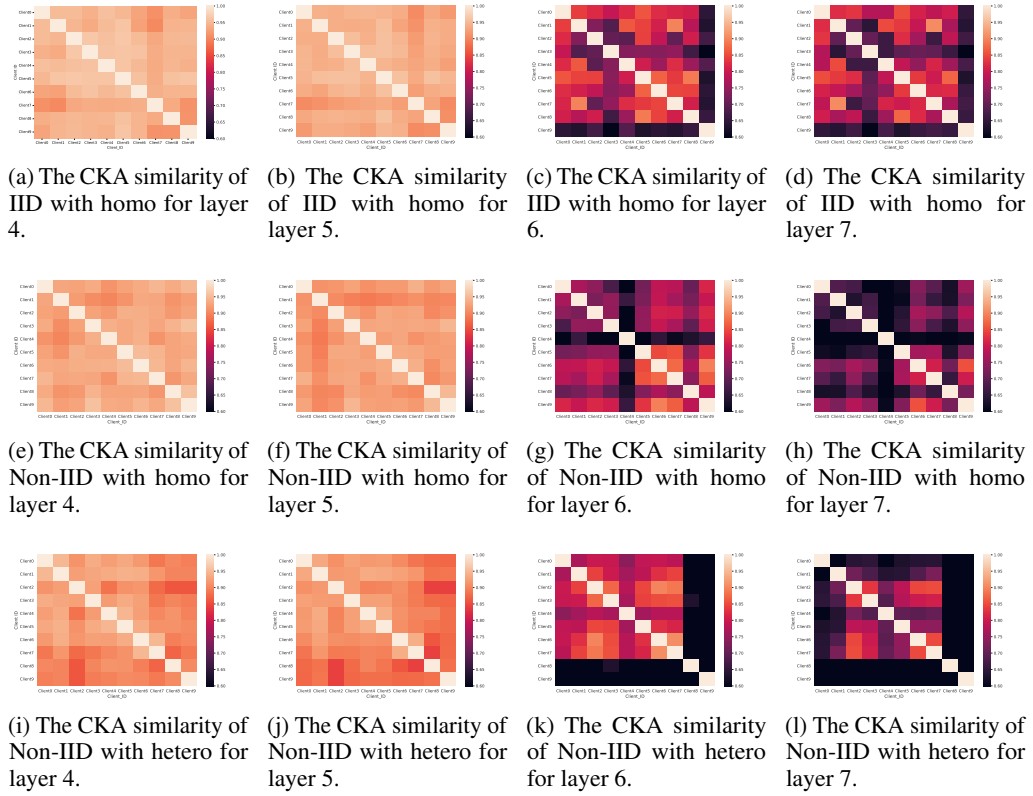

(a) The CKA similarity of IID with homo for layer 4.

(b) The CKA similarity of IID with homo for layer 5.

(c) The CKA similarity of IID with homo for layer 6.

(d) The CKA similarity of IID with homo for layer 7.

(e) The CKA similarity of Non-IID with homo for layer 4.

(f) The CKA similarity of Non-IID with homo for layer 5.

(g) The CKA similarity of Non-IID with homo for layer 6.

(h) The CKA similarity of Non-IID with homo for layer 7.

(i) The CKA similarity of Non-IID with hetero for layer 4.

(j) The CKA similarity of Non-IID with hetero for layer 5.

(k) The CKA similarity of Non-IID with hetero for layer 6.

(l) The CKA similarity of Non-IID with hetero for layer 7.

Figure 18: The CKA similarity of IID with homo, Non-IID with homo and Non-IID with hetero for ViTs.

## H MORE DETAILS OF THE EXPERIMENTS

### H.1 PROCEDURE FOR INCO AGGREGATION

The pseudo-codes for InCo Aggregation are shown in Algorithm 1. InCo Aggregation is operated in a server model, indicating that the methods focused on the client can be aligned with InCo Aggregation, as shown in our experiments.

---

**Algorithm 1** InCo Aggregation (InCoAvg as the example)

**Require:** Dataset $D_k, k \in \{1, ..., K\}$, $K$ clients, and their weights $w_1, ..., w_K$.
**Ensure:** Weights for all clients $w_1, ..., w_K$.
1: **Server process:**
2: **while** *not converge* **do**
3:   Receives $g_{w_i}^t$ from the sampled client.
4:   Parameter aggregation for $g_{w_i}^t$.
5:   **for** *each layer $l_k$ in the server model* **do**
6:     **if** $l_k$ *needs cross-layer gradients* **then**
7:       $g_{l_k}^t{}', g_{l_0}^t{}' \leftarrow$Normalizes $g_{l_k}^t$ and $g_{l_0}^t$.
8:       $\theta^t, \alpha, \beta$ from Theorem 3.1.
9:       $g_{l_k}^{t+1} = \frac{(g_{l_k}^t{}' - \theta^t g_{l_0}^t{}') \times (||g_{l_k}^t|| + ||g_{l_0}^t||)}{2}$
10:    **else**
11:      $g_{l_k}^{t+1} = g_{l_k}^t$
12:    **end if**
13:    $w_{l_k}^{t+1} = w_{l_k}^t + g_{l_k}^{t+1}$
14:  **end for**
15:  Sends the updated $w_i^{t+1}$ to sampled clients.
16: **end while**
17: **Client processes:**
18: **while** *random clients $i, i \in 1, ..., K$* **do**
19:   Receives model weights $w_i^{t-1}$.
20:   Updates client models $w_i^{t-1}$ to $w_i^t$.
21:   Sends $g_{w_i}^t = w_i^t - w_i^{t-1}$ to the server.
22: **end while**

---

Table 5: Test accuracy of 100 clients and sample ratio 0.1. We shade in gray the methods that are combined with our proposed method, InCo Aggregation. We show the error bars for InCo Aggregation in this table.

| Base | Methods | Fashion-MNIST | | SVHN | | CIFAR10 | | CINIC10 | |
|---|---|---|---|---|---|---|---|---|---|
| | | $\alpha = 0.5$ | $\alpha = 1.0$ | $\alpha = 0.5$ | $\alpha = 1.0$ | $\alpha = 0.5$ | $\alpha = 1.0$ | $\alpha = 0.5$ | $\alpha = 1.0$ |
| ResNet | FedAvg | 86.7±1.0 | 87.7±0.6 | 74.8±3.2 | 81.6±2.5 | 52.3±3.4 | 61.3±3.2 | 43.1±2.7 | 49.2±3.1 |
| | FedProx | 75.1±1.8 | 76.6±1.5 | 32.0±2.8 | 43.7±2.9 | 19.2±2.2 | 23.4±2.4 | 17.4±1.7 | 19.8±1.4 |
| | Scaffold | 87.9±0.5 | 88.0±0.3 | 76.3±3.4 | 82.4±3.1 | 54.3±3.6 | 61.8±3.0 | 43.5±2.4 | 49.4±3.1 |
| | FedNova | 12.7±0.2 | 15.6±0.2 | 13.4±0.4 | 15.3±0.3 | 10.4±0.3 | 14.3±0.2 | 12.0±0.3 | 14.0±0.2 |
| | MOON | 87.7±0.4 | 87.5±0.3 | 72.8±4.3 | 81.2±3.2 | 47.2±2.7 | 58.8±2.6 | 40.8±2.1 | 49.2±1.9 |
| | InCoAvg | **90.2±1.2** | 88.4±1.8 | 87.6±2.8 | 89.0±2.6 | 67.8±3.2 | 70.7±3.4 | 53.0±3.2 | 57.5±3.3 |
| | InCoProx | 88.8±2.3 | 86.4±3.2 | **89.0±1.3** | **90.8±1.2** | **74.5±2.3** | **76.8±1.8** | **59.1±3.2** | **62.5±2.4** |
| | InCoScaffold | 88.3±1.4 | **90.1±1.2** | 85.4±2.4 | 87.8±3.5 | 67.3±3.6 | 73.8±2.9 | 53.5±3.3 | 61.7±3.0 |
| | InCoNova | 86.6±1.4 | 87.4±1.3 | 86.4±2.5 | 88.4±1.8 | 62.8±3.9 | 69.7±4.2 | 48.0±2.7 | 54.1±1.7 |
| | InCoMOON | 89.1±1.3 | 89.5±1.2 | 85.6±3.8 | 89.3±2.0 | 68.2±3.1 | 71.8±2.3 | 54.3±3.0 | 57.6±2.7 |
| ViT | FedAvg | 92.0±0.7 | 91.9±0.5 | 92.4±0.9 | 93.9±0.8 | 93.7±1.0 | 94.2±0.8 | 83.8±1.4 | 85.1±0.9 |
| | FedProx | 89.8±0.5 | 89.7±0.5 | 71.4±3.8 | 81.1±2.9 | 82.6±3.3 | 84.7±2.3 | 67.8±2.8 | 71.3±3.0 |
| | Scaffold | 92.0±0.4 | 92.0±0.5 | 92.2±0.8 | 93.8±0.6 | 93.5±0.7 | 94.5±0.5 | 83.3±1.6 | 85.5±1.2 |
| | FedNova | 70.3±0.5 | 76.7±0.4 | 27.4±0.4 | 49.8±0.5 | 30.7±0.3 | 54.4±0.5 | 31.6±1.5 | 50.7±1.3 |
| | MOON | 92.1±0.3 | 92.1±0.2 | 92.5±1.2 | 93.9±0.9 | 93.6±0.8 | 94.6±0.3 | 84.3±1.6 | 85.3±1.2 |
| | InCoAvg | 93.0±0.6 | 93.1±0.5 | 94.2±0.6 | 95.0±0.4 | 94.6±0.7 | 95.0±0.6 | 85.9±1.9 | 86.8±1.3 |
| | InCoProx | 92.6±0.3 | 92.5±0.3 | 93.9±0.7 | 94.4±0.6 | 94.0±1.0 | 94.8±0.7 | 85.1±1.4 | 86.0±0.8 |
| | InCoScaffold | 92.9±0.3 | 93.0±0.2 | 94.0±1.1 | 94.8±0.6 | 94.6±0.5 | 95.0±0.2 | 85.7±1.3 | 86.5±1.1 |
| | InCoNova | **93.1±0.3** | **93.6±0.3** | 94.7±0.9 | **95.6±0.5** | **94.8±0.4** | **95.7±0.3** | **86.2±1.8** | **88.2±1.0** |
| | InCoMOON | 92.8±0.5 | 93.0±0.3 | **94.7±0.8** | 95.1±0.5 | 94.2±0.8 | 95.1±0.5 | 86.0±0.9 | 86.8±1.3 |

## H.2 DATASETS

We conduct experiments on Fashion-MNIST, SVHN, CIFAR-10, and CINIC-10. CINIC-10 is a dataset of the mix of CIFAR-10 and ImageNet within ten classes. We use $3\times224\times224$ with the ViT models and $3\times32\times32$ with the ResNet models for all datasets.

## H.3 HYPER-PARAMETERS

We deploy stage splitting for ResNets and obtain five sub-models, which can be recognized as ResNet10, ResNet14, ResNet18, ResNet22, and ResNet26. For the pre-trained ViT models, we employ layer splitting and obtain five sub-models, which are ViT-S/8, ViT-S/9, ViT-S/10, ViT-S/11, and ViT-S/12 from the PyTorch Image Models (timm)[7]. Our implementations of FedAvg, FedProx, FedNova, Scaffold and MOON are referred to (Li et al., 2022). We use Adam optimizer with a learning rate of 0.001, $\beta_1 = 0.9$ and $\beta_2 = 0.999$, default parameter settings for all methods of ResNets. The local training epochs are fixed to 5. The batch size is 64 for all experiments. Furthermore, the global communication rounds are 500 for ResNets, and 200 for ViTs for all datasets. Global communication rounds for MOON and InCoMOON are 100 to prevent the extreme overfitting in Fashion-MNIST. The hyper-parameter $\frac{\mu}{2}$ for **FedProx** and **InCoProx** is 0.05 for ViTs and ResNets. We conduct our experiments with 4 NVIDIA GeForce RTX 3090s. All baselines and their InCo extensions are conducted in the same hyper-parameters. The settings of hetero splitting for ScaleFL followed the source codes.[8]

## H.4 MODEL SIZES

We demonstrate the model sizes for each client model in Table 7, Table 8, and Table 9.

## H.5 ERROR BARS OF INCO AGGREGATION

We illustrate the error bars of InCo Aggregation and the results from model-homogeneous baselines (not use stage or layer splitting in the model heterogeneous environment) in Table 5. For the model-heterogeneity methods, we demonstrate the error bars of InCo Aggregation in Table 6. These results show the stability of InCo Aggregation. In all cases of ResNets and many cases of ViTs, the worst

---

[7]https://github.com/rwightman/pytorch-image-models
[8]https://github.com/git-disl/scale-fl

Table 6: Test accuracy of model-heterogeneity methods with 100 clients and sample ratio 0.1. We shade in gray the methods that are combined with our proposed method, InCo Aggregation. We show the error bars for InCo Aggregation in this table.

| Base | Splitting | Methods | Fashion-MNIST | | SVHN | | CIFAR10 | |
| --- | --- | --- | --- | --- | --- | --- | --- | --- |
| | | | $\alpha = 0.5$ | $\alpha = 1.0$ | $\alpha = 0.5$ | $\alpha = 1.0$ | $\alpha = 0.5$ | $\alpha = 1.0$ |
| ResNet | Hetero | HeteroFL | 88.9±1.0 | 89.7±0.7 | 90.5±1.6 | 92.2±1.3 | 65.2±3.2 | 68.4±3.6 |
| | | +InCo | 90.0±1.2 | 90.4±1.1 | 92.1±1.0 | 93.5±1.5 | 68.2±3.8 | 71.2±3.4 |
| | Stage | InclusiveFL | 89.1±1.1 | 89.8±1.0 | 88.6±2.0 | 90.0±2.2 | 65.7±3.5 | 68.4±3.3 |
| | | +InCo | 90.1±1.5 | 90.5±1.3 | 90.6±1.7 | 90.9±1.9 | 69.1±2.8 | 72.3±3.1 |
| | Hetero | FedRolex | 88.2±1.0 | 90.2±0.8 | 90.9±1.3 | 91.6±1.7 | 64.7±4.1 | 72.3±3.0 |
| | | +InCo | 90.4±1.4 | 91.3±1.1 | 92.8±1.5 | 93.4±1.6 | 67.9±2.9 | 75.6±2.6 |
| | Hetero | ScaleFL | 90.9±0.5 | 91.0±0.4 | 92.6±1.0 | 92.9±0.9 | 71.1±2.9 | 74.7±3.1 |
| | | +InCo | 91.5±1.0 | 91.7±1.1 | 93.4±0.9 | 93.6±0.9 | 73.8±3.2 | 76.1±2.6 |
| | N/A | AllSmall | 83.5±1.7 | 84.0±1.7 | 72.1±3.5 | 81.0±2.9 | 39.2±2.0 | 44.9±2.3 |
| | N/A | AllLarge | 91.8±0.5 | 92.5±0.8 | 93.4±0.8 | 93.8±0.5 | 79.6±2.9 | 82.5±1.0 |

Table 7: Model parameters for different architectures of ResNets (Stage splitting).

| Sizes | ResNets (Stage splitting) | | | | |
| --- | --- | --- | --- | --- | --- |
| | ResNet10 | ResNet14 | ResNet18 | ResNet22 | ResNet26 |
| Params | 4.91M (×0.281) | 10.81M (×0.619) | 11.18M (×0.641) | 17.08M (×0.979) | 17.45M (×1) |

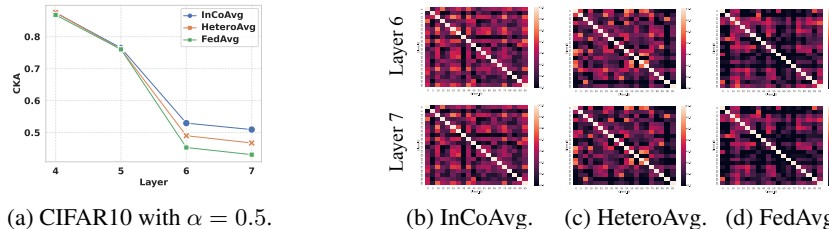

(a) CIFAR10 with $\alpha = 0.5$.      (b) InCoAvg.    (c) HeteroAvg.    (d) FedAvg.

Figure 19: CKA layer similarity and Heatmaps of ViTs. (a): The layer similarity of different methods. (b) to (d): Heatmaps for different methods in layer 6 and layer 7.

results of InCo Aggregation are better than the Averaging Aggregation, demonstrating the efficacy of InCo Aggregation.

### H.6 DIFFERENCES BETWEEN ADDING NOISES AND INCO GRADIENTS.

The convergence speed of InCo gradients surpasses that of the other two methods, as illustrated in Figure 20a. Table 20c demonstrates that InCo gradients outperform other methods across different datasets. The primary distinction between InCo gradients and adding noises lies in their ability to determine the precise gradient modification for each node in the model. Gaussian noise lacks the capability to specify the exact modification required for each node, leading to less controlled and targeted adjustments. This is evident in Figure 20b, where the Frobenius norm of noises is larger and exhibits greater instability compared to InCo gradients.

### H.7 LAYER SIMILARITY AND HEATMAPS OF VITS

Figure 19 illustrates the layer similarity of the last four layers, along with the corresponding heatmaps for Layer 6 and Layer 7. Furthermore, Figure 19a demonstrates that InCo Aggregation significantly enhances the layer similarity, validating the initial motivation behind our proposed method. Additionally, since the disparity in layer similarity between Layer 6 and Layer 7 is minimal, the heatmaps for these layers do not exhibit significant differences, as depicted in Figure 19b through Figure 19d.

Table 8: Model parameters for different architectures of ViTs (Layer splitting).

| Sizes | ViTs (Layer splitting) | | | | |
|---|---|---|---|---|---|
| | ViT-S/8 | ViT-S/9 | ViT-S/10 | ViT-S/11 | ViT-S/12 |
| Params | 14.57M ($\times$0.672) | 16.34M ($\times$0.754) | 18.12M ($\times$0.836) | 19.90M ($\times$0.912) | 21.67M ($\times$1) |

Table 9: Model parameters for different architectures of ResNets (Heterogeneous splitting).

| Sizes | ResNets (Hetero splitting) | | | | |
|---|---|---|---|---|---|
| | $\frac{1}{16}$ | $\frac{1}{8}$ | $\frac{1}{4}$ | $\frac{1}{2}$ | ResNet26 |
| Params | 0.07M ($\times$0.004) | 0.28M ($\times$0.016) | 1.10M ($\times$0.06) | 4.37 ($\times$0.25) | 17.45M ($\times$1) |

### H.8 MORE ABLATION STUDIES AND ROBUSTNESS ANALYSIS.

We conduct additional experiments on different baselines to demonstrate the effectiveness of InCo Aggregation. Figure 21 to Figure 23 present the results of the ablation study for FedProx, FedNova, and Scaffold, incorporating InCo Aggregation. These results highlight the efficacy of InCo Aggregation across different baselines. Additionally, Figure 24 and Figure 25 illustrate the robustness analysis for FedProx and Scaffold. In Figure 24 and Figure 25, InCoProx and InCoScaffold consistently obtains the best performances across all settings. These experiments provide further evidence of the efficiency of InCo Aggregation.

## I LIMITATIONS AND FUTURE DIRECTIONS

The objective of this study is to expand the capabilities of model-homogeneous methods to effectively handle model-heterogeneous FL environments. However, the analysis of layer similarity reveals that the smallest models do not derive substantial benefits from InCo Aggregation, implying the limited extensions for these smallest models. Exploring methods to enhance the performance of the smallest models warrants further investigation. Furthermore, our research mainly focuses on image classification tasks, specifically CNN models (ResNets) and Transformers (ViTs). However, it is imperative to validate our conclusions in the context of language tasks, and other model architectures such as LSTM Hochreiter & Schmidhuber (1997). Additionally, it is important to consider that the participating clients in the training process may have different model architectures. For example, some clients may employ CNN models, while others may use Vision Transformers (ViTs). We believe that it is worth extending this work to encompass a wider range of tasks and diverse model architectures that hold great value and potential for future research.

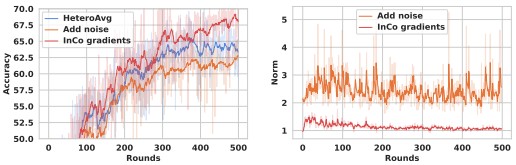

| Methods | FashionMNIST | | SVHN | | CIFAR10 | |
|---|---|---|---|---|---|---|
| | 0.5 | 1.0 | 0.5 | 1.0 | 0.5 | 1.0 |
| HeteroAvg | 87.8 | 86.0 | 85.1 | 86.9 | 64.8 | 66.7 |
| Add noises | 87.0 | 86.7 | 83.3 | 86.2 | 62.5 | 64.9 |
| InCo gradients | **90.2** | **88.4** | **87.6** | **89.0** | **67.8** | **70.7** |

(a) Convergence speeds.  (b) Frobenius norm.  (c) Accuracy for different datasets.

Figure 20: Convergence speeds, Frobenius norm of adding noise and InCo gradients (InCoAvg), and accuracy results for different datasets. (a): Convergence speeds of HeteroAvg, adding noise and InCo gradients. (b): Frobenius norm of noise and InCo gradients. (c): Accuracy for different datasets.

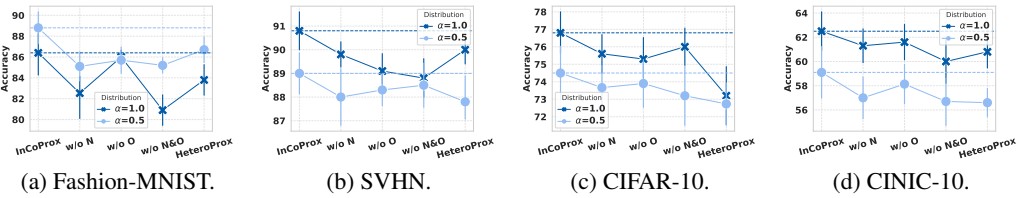

(a) Fashion-MNIST.  (b) SVHN.  (c) CIFAR-10.  (d) CINIC-10.

Figure 21: Ablation studies for InCo Aggregation for FedProx. The federated settings are the same as Table 2.

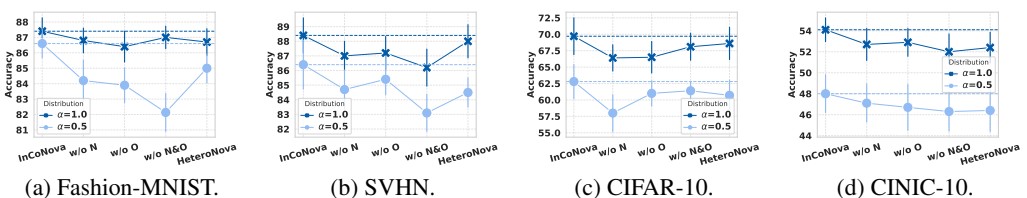

(a) Fashion-MNIST.  (b) SVHN.  (c) CIFAR-10.  (d) CINIC-10.

Figure 22: Ablation studies for InCo Aggregation for FedNova. The federated settings are the same as Table 2.

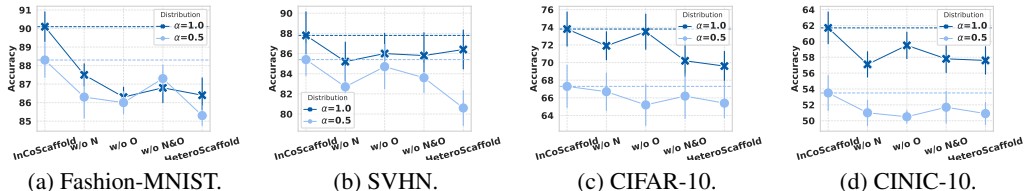

(a) Fashion-MNIST.  (b) SVHN.  (c) CIFAR-10.  (d) CINIC-10.

Figure 23: Ablation studies for InCo Aggregation for Scaffold. The federated settings are the same as Table 2.

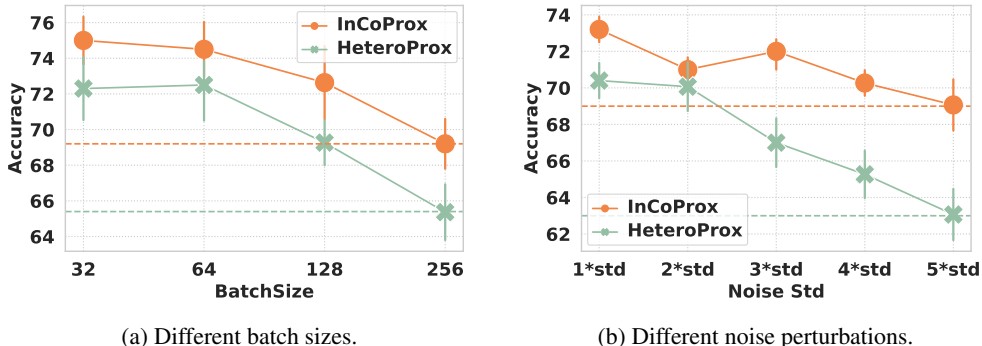

(a) Different batch sizes.  (b) Different noise perturbations.

Figure 24: Robustness analysis for InCo Aggregation for FedProx in CIFAR-10.

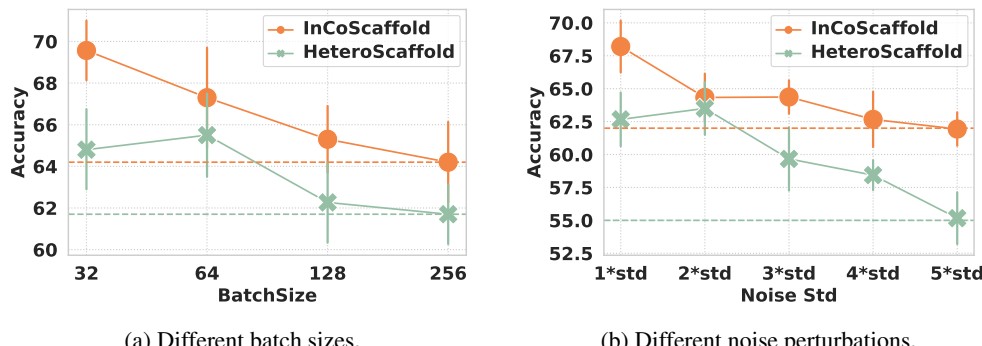

(a) Different batch sizes.

(b) Different noise perturbations.

Figure 25: Robustness analysis for InCo Aggregation for Scaffold in CIFAR-10.

