# OpenReview forum: "Internal Cross-layer Gradients for Extending Homogeneity to Heterogeneity in Federated Learning"
_ICLR.cc/2024/Conference — ICLR 2024 poster_

### Official Review · Reviewer_ky42 · 2023-10-19

**Soundness:** 3 good
**Presentation:** 3 good
**Contribution:** 3 good
**Rating:** 8
**Confidence:** 2

**Summary:**

this paper discover three interesting observations based on the exploration between homogeneous and heterogeneous FL settings. Then the authors propose InCo Aggregation methods, inpried by these observation and demonstrate the proposed method can be tailored to accommodate model-homogeneous FL methods and achieve better performance.

**Strengths:**

The paper is well-written and easily understandable, effectively communicating the research in a clear and coherent manner.

The discovered observations are interesting and valuable for future research, providing a foundation for further investigations and potential advancements in the field.

The experiments conducted in the paper are sufficient, with an appropriate and comprehensive setup that collects relevant data to support the claims and conclusions.

**Weaknesses:**

1. CKA is an important metric in this paper. The authors should explain it in more details.
2. Model splitting is proposed to facilitate model heterogeneity. What if the layer-wise gradient sizes of different models are not the same, how do you conduct cross-layer gradients mergence?

**Questions:**

1. where is CKA from in Fig. 1 (c). Is it the average value of all stages or deep/shallow stage?
2. Can you provide a more detailed comparison of InCo Aggregation with other state-of-the-art methods for handling system heterogeneity in FL? How does InCo Aggregation compare in terms of performance, communication overhead, and computational complexity?
3. The paper does not provide a detailed analysis of the computational and communication overhead of InCo Aggregation, which could be a significant factor in large-scale FL applications. This limits the practical applicability of the proposed approach.

**Details Of Ethics Concerns:**

null

---

> ### Author Response · Authors · 2023-11-19
> **Response to Reviewer ky42 [1/1]**
>
> Thank you for your valuable and constructive review. We now address your main concerns as follows.
>
> ### Weakness
>
> > **W1.** CKA is an important metric in this paper. The authors should explain it in more details.
>
> **Answer:**
> - Thank you for your comment. We have updated the detailed introduction of CKA in Appendix A to ensure clarity and accuracy in describing the motivation of our method.
>
> > **W2.** Model splitting is proposed to facilitate model heterogeneity. What if the layer-wise gradient sizes of different models are not the same, how do you conduct cross-layer gradients mergence?
>
> **Answer:**
> - Thank you for your suggestion. Currently, InCo Aggregation is focused on addressing the issue of model heterogeneity within the same family of backbones with varying depths and widths. It is not suitable for handling model heterogeneity across completely different model families, such as a combination of CNNs and ViTs. This aspect requires further research. We have updated and discussed these limitations of InCo Aggregation in Appendix I.
> - However, regarding the problem of model heterogeneity across completely different architectures, we have identified two potential directions that could be explored in the context of InCo Aggregation. The first direction involves leveraging public datasets to perform knowledge distillation, where the knowledge from different model architectures is distilled onto one or several server models with the same architecture (more server models can store more knowledge). InCo Aggregation can then be performed on these server models to connect clients with different model architectures. This direction is similar to the approach proposed in FedDF [1].
> - The second method involves utilizing hypernetworks to generate parameters for corresponding layers. Different model architectures can be connected through hypernetworks, similar to how different input dimensions can be transformed into the same dimensions of layer parameters through hypernetworks. Similar to the previous direction, multiple server models can be generated using hypernetworks, followed by InCo Aggregation on the generated server models. Currently, there are several papers focusing on personalized federated learning that utilize hypernetworks for generating weights, such as [2], [3], and [4]. However, we have not yet come across any methods using hypernetworks for addressing model heterogeneity in federated learning.
> - These are preliminary ideas to make InCo Aggregation applicable to model heterogeneity across completely different backbones. However, further in-depth research and experimental analysis are required to explore these directions fully.
>
> ### Questions
>
> > **Q1.** Where is CKA from in Fig. 1 (c). Is it the average value of all stages or deep/shallow stage?
>
> **Answer:**
> - Thank you for your comment. In order to obtain a clearer observation of the relationship between CKA and accuracy, we utilized the average value of stage 0. However, it is worth noting that similar relationships exist for other stages as well. To provide a comprehensive analysis, we update the relationships between CKA and accuracy for other stages in Figure 12c and 12d, as well as Appendix G.2. The results show that the positive correlation remains consistent cross different stages. This consistency indicates that this relationship is reliable and stable throughout the various stages.
>
> > **Q2&Q3.** Computational and communication overhead of InCo Aggregation.
>
> **Answer:**
>
> - Thank you for your suggestion. We have updated a comparison of communication overhead and computational resource utilization (FLOPs) in Table 3. We also describe these comparison as follows,
>
> |                 | HeteroFL | +InCo | InclusiveFL | +InCo | FedRolex | +InCo | ScaleFL |  +InCo   |
> | --------------- | -------- | ----- | ----------- | ----- | -------- | ----- | ------- | --- |
> | Comm. Overheads | 4.6M     | 4.6M  | 12.3M       | 12.3M | 4.6M     | 4.6M  | 9.5M    |  9.5M   |
> | FLOPs           | 33.4M    | 33.8M | 75.2M       | 75.6M | 33.4M    | 33.8M | 51.9M   |  52.3M   |
>
> - The computational load of InCo Aggregation arises from the computation of Equation (2). It is noteworthy that InCo Aggregation does not impose any additional communication overhead since there is no requirement for clients to transmit supplementary information. Moreover, InCo Aggregation does not place any burdens on the computation resources of the clients, as the operations involving cross-layer gradients and the gradient alleviation theorem are conducted exclusively on the server side.
>
>
> [1] Ensemble distillation for robust model fusion in federated learning. NeurIPS 2020.
>
> [2] Personalized federated learning using hypernetworks. ICML 2021.
>
> [3] Layer-wised Model Aggregation for Personalized Federated Learning. CVPR 2022.
>
> [4] Efficient model personalization in federated learning via client-specific prompt generation. ICCV 2023.

---

### Official Review · Reviewer_ibVy · 2023-10-29

**Soundness:** 2 fair
**Presentation:** 2 fair
**Contribution:** 2 fair
**Rating:** 5
**Confidence:** 4

**Summary:**

The paper tackles the challenge of heterogeneous models within federated learning, uncovering a notable pattern where shallow layers exhibit similar gradient distributions, in contrast to the disparate distributions in deeper layers. The authors further observed that higher gradient similarity corresponds to improved accuracy. Capitalizing on these insights, a novel aggregation method is proposed and substantiated through comprehensive experiments, demonstrating the efficacy of the proposed approach in navigating model heterogeneity.

**Strengths:**

1. The author presents intriguing findings regarding the relationship between layer similarity and model performance.
2. By refining the direction of deep layers, which exhibit lower gradient similarity compared to shallow layers, the author enhances model performance.
3. The proposed method demonstrates versatility, adaptable to various Federated Learning (FL) schemas.
4. The effectiveness of the proposed method is substantiated through extensive experimentation.

**Weaknesses:**

1. The utilization of a model with merely three convolutional layers is unconvincing; larger models should be employed in primary experiments to validate the findings.
2. The manuscript could benefit from a more coherent narrative, including background on previous works, an introduction to the CAK similarity metric, and a discussion on why the proposed method outperforms state-of-the-art (SOTA) methods, especially in handling model heterogeneity.
3. Figure 6 lacks clarity; indicating the positions of client and global optima could elucidate the depicted concepts.
4. It is imperative to delineate the problem definition and notations before introducing the method, ensuring a logical flow and better comprehension.
5. The manuscript does not adequately explain how cross-layer gradient adjustments ameliorate the effects of model heterogeneity.

**Questions:**

1. I wonder if the proposed approach can be applied to complex models.
2. I wonder how cross-layer gradient adjustments ameliorate the effects of model heterogeneity.

---

> ### Author Response · Authors · 2023-11-19
> **Response to Reviewer ibVy [1/3]**
>
> Thank you for your valuable and constructive review. We now address your main concerns as follows.
>
> ### Weakness
>
> > **W1.** The utilization of a model with merely three convolutional layers is unconvincing; larger models should be employed in primary experiments to validate the findings.
>
> **Answer:**
> - Thank you for pointint out this issue, and sorry for the confusion. In our primary experiments, we do not analyze models with only three convolutional layers. In the model homogeneity (homo) setting, we use ResNet18 and ViT-S/12 as client models. In the model heterogeneity setting, we employe five ResNet models obtained by stage splitting from ResNet26, resulting in ResNet10, ResNet14, ResNet18, ResNet22, and ResNet26. Additionally, we use five ViT models, namely ViT-S/8, ViT-S/9, ViT-S/10, ViT-S/11, and ViT-S/12, obtained by layer splitting from ViT-S/12. The reason for focusing on three-layer convolutional layers in Figure 2 and 3 is that we analyzed layers within the same shape in the stages of ResNet. However, ResNet typically contains 3-4 stages, so the analyzed models are not limited to only three convolutional layers. In our primary experiments, we specifically analyzed Stage 2 and Stage 3 in ResNet18.
> - Furthermore, we employed ViT-S models ranging from 8 blocks to 12 blocks, and ViT models are considered relatively complex within the context of federated learning. In Appendix F.2, we added an analysis of blocks in ViTs, demonstrating that blocks in ViTs exhibit similar relationships between gradient distribution and similarity as stages in ResNet. This confirms the suitability of applying InCo aggregation to ViTs as well. Moreover, our experimental results demonstrate the applicability of InCo aggregation to ViTs.
> - Finally, we provide the model size for the architecture we used, which are shown below and are also included in Table 7, 8, and 9 in Appendix H.4:
>
> |      | ViT-S/8 | ViT-S/9 | ViT-S/10 | ViT-S/11 | ViT-S/12    |
> | ---- | ------- | ------- | -------- | -------- | --- |
> | Params | 14.57M($\times$ 0.67)    | 16.34M ($\times$ 0.754) | 18.12M ($\times$ 0.836) | 19.90M ($\times$ 0.912) | 21.67M ($\times$ 1) |
>
> |      | ResNet10 | ResNet14 | ResNet18 | ResNet22 | ResNet26 |
> | ---- | ------- | ------- | -------- | -------- | --- |
> | Params | 4.91M ($\times$ 0.281) | 10.81M ($\times$ 0.619) | 11.18M ($\times$ 0.641) | 17.08M ($\times$ 0.979) | 17.45M ($\times$ 1)  |
>
> > **W2.** The manuscript could benefit from a more coherent narrative, including background on previous works, an introduction to the CAK similarity metric, and a discussion on why the proposed method outperforms state-of-the-art (SOTA) methods, especially in handling model heterogeneity.
>
> **Answer:**
> - Thank you for your suggestion. Due to the page limitation, we put the most crucial content in the main pages of the paper. The background of previous work is covered in Appendix E to provide a comprehensive understanding of the research context. Additionally, we have updated the detailed introduction of CKA in Appendix A to ensure clarity and accuracy in describing the motivation of our method.
> - For a discussion on why the proposed method outperforms SOTA methods，we provide comprehensive analyses of how each component of InCo Aggregation leads to improvements in Section 5.4. Furthermore, we provide a comprehensive explanation of the underlying factors that enable InCo Aggregation to achieve superior results as follows.
> - The improvements introduced by InCo Aggregation originate from two crucial aspects:
>     - (1) The utilization of internal cross-layer gradients facilitates the propagation of generalized knowledge from shallow layers to deep layers, enabling the deep layers to acquire additional knowledge. Consequently, the model becomes more generalized and avoids biases towards local datasets.
>     - (2) The Divergence Alleviation Theorem (Theorem 3.1) alleviates conflicts between cross-layer gradients and gradients from deep layers.
> - In Section 5.4, Figures 9d-9f provide vision representations of how InCoAvg's features exhibit a higher degree of generalization compared to FedAvg and HeteroAvg, which often display biased phenomena stemming from model heterogeneity. Furthermore, Figure 10a quantitatively describes the efficacy of InCoAvg in promoting model generalization using the CKA similarity measure. Specifically, InCoAvg exhibits higher CKA similarity in deep layers compared to FedAvg and HeteroAvg. The results obtained from Figures 9d-9f and 10a validate that the utilization of internal cross-layer gradients effectively facilitates the transfer of generalized knowledge from shallow layers to deep layers. This transfer mechanism helps alleviate the bias issue within models, consequently improving their overall performance.
>
> [Our response continues.]

---

> > ### Author Response · Authors · 2023-11-19
> > **Response to Reviewer ibVy [2/3]**
> >
> > - Figures 9a-9c and Table 4 elucidate the two most significant parameters within the Divergence Alleviation Theorem: $\theta$, which leverages the weight of $g_0$, and $\beta$, which indicates whether $g_0$ and $g_k$ align in the same direction. Figure 9a illustrates how this theorem automatically adjusts different $g_0$ values according to the layers, eliminating the need for heavily manual parameter tuning. Additionally, the ablation study demonstrates that, under unchanged $\theta$ conditions, the algorithm exhibits lower accuracy compared to InCo. Figures 9b, 9c, and Table 4 further demonstrate that the Divergence Alleviation Theorem effectively mitigates conflicts between cross-layer gradients and gradients from deep layers. This is evident from the noticeable improvement in the proportion of $beta>0$ in Table 4, throughout the entire training process, signifying a reduction in the conflicts between cross-layer gradients and gradients from deep layers. Ultimately, this leads to an improvement in the performance of the client model.
> >
> > > **W3.** Figure 6 lacks clarity; indicating the positions of client and global optima could elucidate the depicted concepts.
> >
> > **Answer:**
> > - Thank you for your comment. We have updated Figure 6 by adding a contour plot of the gradients, as well as client and global optima. These additions enhance the clarity of Figure 6, providing a more comprehensive visual representation of our methods.
> >
> > > **W4.** It is imperative to delineate the problem definition and notations before introducing the method, ensuring a logical flow and better comprehension.
> >
> > **Answer:**
> > - Thank you for your suggestion. To provide a better understanding of the our problem, we have updated more comprehensive problem definition and notations in our paper. Specifically, we consider a set of physical resources denoted as $\\{R_i\\}\_{i=1}^n$, where $R_i$ represents the available resources of client $i$. For each local client model $\\{M_i\\}\_{i=1}^n$, the resource requirement $R(M_i)$ must be smaller than or equal to the available resources of client $i$, i.e. $\\{R(M_i)\leq R_i\\}\_{i=1}^n$. To satisfy this constraint, the client models have varying sizes and architectures. Let $w_i$ denote the weights of the client model $M_i$, and $f(x,w_i)$ represent the forward function of model $M_i$ with input $x$. Moreover, each client has a local dataset $D_i=\\{(x_{k,i}, y_{k,i})|k\in \\{1,2,...,|D_i|\\}\\}$, where $|D_i|$ signifies the size of a dataset $D_i$. The loss function $l_i$ of client $i$ is shown as follows,
> > \begin{equation}
> > \begin{aligned}
> >  \min_w\  l_i(w_i)=\frac{1}{|D_i|}\sum_{k=1}^{|D_i|}{l_{CSE}(f(x_k;w_i), y_k)},
> > \end{aligned}
> > \end{equation}
> >   where $l_{CSE}$ is a cross-entropy function. Moreover, if we denote $K=\sum_{i=1}^n |D_i|$ as the total size of all local datasets, the global optimization problem is,
> > \begin{equation}
> > \begin{aligned}
> >  \min_{w_1, w_2, ..., w_n}\ L(w_1, ..., w_n)=\sum_{i=1}^n\frac{|D_i|}{K}{l_i(w_i)},
> > \end{aligned}
> > \end{equation}
> >   where the optimized model weights $\\{w_1, w_2, ..., w_n\\}$ are the parameters from $\{M_i\}_{i=1}^n$. In our case, $\\{w_1, w_2, ..., w_n\\}$ are split from the server model weight $w_s$ from the server model $M_s$. The goal of our paper is to propose a method that can effectively optimize this equation.
> >
> > - Moreover, we have revised the definitions for system heterogeneity and model heterogeneity in the introduction section to ensure a logical flow. Furthermore, we have updated the additional problem formulation in Appendix F to provide a more comprehensive and improved understanding of our problem. These updates aim to enhance the clarity and coherence of our paper for readers.
> >
> > [Our response continues.]

---

> > > ### Author Response · Authors · 2023-11-19
> > > **Response to Reviewer ibVy [3/3]**
> > >
> > > > **W5.** The manuscript does not adequately explain how cross-layer gradient adjustments ameliorate the effects of model heterogeneity.
> > >
> > > **Answer:**
> > > - Thank you for your comment. Based on our case study, it has been observed that under conditions of model heterogeneity, deep layers exhibit lower similarity compared to model homogeneity. This disparity in similarity indicates that deep layers are more biased, consequently leading to a decline in the performance of the client model. Therefore, our objective is to mitigate the bias present in deep layers, thereby enhancing the overall performance of the model.
> > > - To achieve this, we propose the utilization of cross-layer gradients, which facilitate the transfer of generalized knowledge from shallow layers to deep layers. By leveraging cross-layer gradients, we can alleviate the excessive bias observed in deep layers within the context of model heterogeneity. As a result, the adverse effects of model heterogeneity can be mitigated, thereby improving overall model performance.
> > > - Furthermore, Figures 9d-9f provide visual representations that illustrate how cross-layer gradients alleviate the effects of model heterogeneity. It is important to note that in Figure 9d, the separation of features is a result of model heterogeneity rather than data heterogeneity. This is because the features in Figure 9d are clustered based on the model architectures. In Figure 9d, client 0 and client 1 utilize ResNet10 and ResNet14, respectively, both of which are relatively smaller models within our model heterogeneity environment. On the other hand, client 2 employs ResNet26, which is the largest model in our model heterogeneity environment. As a result, in Figure 9d, the features of client 0 and client 1 form one cluster, while the features of client 2 belong to a separate cluster. Additionally, the labels of the three clients are drawn from different distributions ($\alpha=0.5$ from Dirichlet distribution). However, despite the three different data distributions, the features of these three clients do not form three distinct clusters but rather split into two clusters based on the model architectures. Figure 9e also exhibits a similar pattern, although the feature separation is not as obvious as in Figure 9d. Subsequently, in Figure 9f, after applying InCo Aggregation, the feature bias induced by model architectures is eliminated. Even though the architectures of the three clients remain different, their generated features now reside within a single cluster. Therefore, Figures 9d-9f provide a clear illustration that cross-layer gradients enhance the generalization capability of the model, thereby alleviating the bias introduced by model heterogeneity.
> > >
> > > ### Questions
> > >
> > > > **Q1.** I wonder if the proposed approach can be applied to complex models.
> > >
> > > **Answer:**
> > > - Thank you for your question. Yes, in our experiments, we have five ResNet models which are stage splitting from ResNet26, resulting in ResNet10, ResNet14, ResNet18, ResNet22, and ResNet26. Five ViTs models are ViT-S/8, ViT-S/9, ViT-S/10, ViT-S/11, ViT-S/12, the results from the layer splitting of ViT-S/12.
> > > - Moreover, we provide the model size for the architecture we used. Please refer to the response regarding weakness 1.
> > >
> > > > **Q2.** I wonder how cross-layer gradient adjustments ameliorate the effects of model heterogeneity.
> > >
> > > **Answer:**
> > > - Thank you for your question. Please refer to the response regarding weakness 5.

---

> > > > ### Author Response · Authors · 2023-11-22
> > > > **Official Comment by Authors**
> > > >
> > > > Dear Reviewer ibVy,
> > > >
> > > > We sincerely thank you for investing your time in reviewing our work and for providing valuable feedback. Your comments have been truly invaluable in enhancing the clarity and readability of our content. As we approach the end of the discussion period, please feel free to reach out if you have any additional questions or require further information. We are here to assist you with great enthusiasm. If our responses have met your expectations, we would greatly appreciate your acknowledgment of our efforts. Once again, we express our gratitude for your significant contribution to improving our work. We eagerly anticipate receiving your feedback.
> > > >
> > > > Best regards,
> > > >
> > > > The authors

---

### Official Review · Reviewer_e8EV · 2023-10-31

**Soundness:** 4 excellent
**Presentation:** 3 good
**Contribution:** 3 good
**Rating:** 8
**Confidence:** 3

**Summary:**

The paper highlights the positive correlation between client performances and layer similarities in a federated setting, specifically how similarities are higher in shallow layers of the network and steadily decrease as one gets to the deeper layers as well as how smoother gradient distributions correspond to higher layer similarities.
Keeping these ideas in mind, the paper proposes InCo Aggregation, which leverages cross-layer gradient similarities to improve the similarity across clients in deeper layers, without the need for additional communication between clients.
Overall, InCo Aggregation is validated on both model-homogeneous and heterogeneous methods to highlight its impact in improving performances across the board.

**Strengths:**

- The observations on similarity at varying levels of deep networks across clients could potentially establish an observation mechanism for federated settings to analyze the impact of changes across multiple axes like data settings, model heterogeneity, etc.
- Figures 4 and 6 do a good job at conveying concepts surrounding various model splitting methods and gradient divergence, respectively.

**Weaknesses:**

- "System level heterogeneity" is mentioned multiple times and is loosely defined through the early portion of the manuscript (Pg. 1, Paragraph 1). Over the course of reading the paper, one can figure out data and model heterogeneity are the relevant axes along which system level heterogeneity is defined. Defining these ideas earlier and more concisely would allow the reader to contextualize the problem domain and understand how the solution being proposed fits within its scope.
- While the paper discusses the relationship between gradient similarity and performance/accuracy, and gradient similarity with smoother gradients, the key justifications for choosing to use smoother gradients are (a)  lack of a shared database, (b) Features would increase communication overheads, and (c) correlation between similarity of gradients and smoothness. Are there stronger correlations between the level of similarity (actual value) to the peak density or other statistics of the gradients?
- There doesn't seem to be a an explicit definition of "deep" vs. "shallow" layers. Implicitly, within each stage there seem to be shallow and deep layers (Pg. 3, Cross-environment similarity). In general, there are certain experimental settings necessary to fully understand the figures plotted through the course of the manuscript that seem to be missing.
- Given the specific model heterogeneity settings under which InCo Aggregation is applicable, a discussion on how and where it isn't applicable (e.g., Complete model heterogeneity, where models are restricted to be within the same family of backbones) would be useful.

**Questions:**

- Could the authors define the notion of system level heterogeneity, using model and data heterogeneity, earlier and provide connection points to how the proposed method would address these issues. Having these points described earlier would allow the reader to grasp the importance of the observations described later on.
- Could the authors provide gradient plots like in Figs 2 and 3 for Stage 2's layers as well? Drawing a parallels to behavior across Stages would be helpful in establishing the consistency of the observations on smoother gradients and how they relate to similarity values.
- Could the authors provide an explicit definition of which layers can be considered deep vs.shallow? If the nomenclature implicitly defined in "Cross-environment Similarity" is to be maintained, could the authors provide an explanation of whether this pattern carries over to ViTs as well?
- Could the authors discuss further about similarity patterns in ViT's and how this impact the observations and InCo aggregation as a whole?
- Could the authors provide more detail explanations for the exact setups used to generate Fig. 1, 2, and 3?
- Could the authors provide more insight into how Fig. 3 would vary when tested across multiple trials? This could help remove the uncertainty caused by SGD noise.
- Could the authors provide the standard deviation values for InCo-based methods?
- Given the variation in performances in Table 3 and the original values cited under FedROLEX, ScaleFL, etc., could the authors provide a detailed breakdown of how the experimental settings differ from the original works?
- A discussion on how and where it isn't applicable (e.g., Complete model heterogeneity, where models are restricted to be within the same family of backbones or in cases where weight matrices do no align) is critical to understand and apply the proposed method.

---

> ### Author Response · Authors · 2023-11-19
> **Response to Reviewer e8EV [1/4]**
>
> Thank you for your valuable and constructive review. We now address your main concerns as follows.
>
> Weaknesses:
>
> > **W1.** "System level heterogeneity" is mentioned multiple times and is loosely defined through the early portion of the manuscript (Pg. 1, Paragraph 1).
>
> **Answer:**
> - Thank you for your valuable suggestions. To provide a better understanding of the problem domain, we have updated more comprehensive descriptions and definitions of system heterogeneity and model heterogeneity. System heterogeneity refers to a set of varying physical resources $\\{R_i\\}\_{i=1}^n$, where $R_i$ denotes the resource of client $i$, a high-level idea of resource that holistically governs the aspects of computation, communication, and storage. Model heterogeneity refers to a set of different local models $\\{M_i\\}\_{i=1}^n$ with $M_i$ being the model of client $i$. Let $R(M)$ denote the resource requirement for the model $M$. Model heterogeneity is a methodology that manages to meet the constraints $\\{R(M_i)\leq R_i\\}_{i=1}^n$. We have updated these descriptions in the introduction section.
> - Additionally, in the case study section, we have clarified the level of data heterogeneity to give a clearer context for the analysis. Moreover, the additional details about the problem formulation in Appendix F provide readers with a more in-depth understanding of the problem domain.
>
> > **W2.** Are there stronger correlations between the level of similarity (actual value) to the peak density or other statistics of the gradients?
>
> **Answer:**
> - Thank you for your question. We have attempted various statistics, including the means, variances, differences and covariances between gradients from the same layer within different environments. However, none of these statistics have exhibited a stronger correlation with the similarity of gradients compared to the smoothness. We have updated these results in Figure 16 and Appendix G.5. Therefore, we select correlation with smoothness to reflect that the shallow layers have higher similarity and smoother gradient distributions. Based on this observation, we propose InCo Aggregation.
> - Furthermore, we have updated an analysis for the correlation between the similarity and smoothness for ViTs in Appendix G.3. This supplementary analysis further supports the observation of a correlation between similarity and smoothness in ViTs as well.
>
> > **W3.** There doesn't seem to be an explicit definition of "deep" vs. "shallow" layers.
>
> **Answer:**
> - Thank you for pointing out this issue. "A deep layer" refers to the deep layer within a stage (block) in our paper, with the exception of Figure 1. We would like to clarify that in Figure 1, the term "deep layer" is indeed the deep layer of the entire model, because Fig 1 is an illustrative example that facilitates the reader's understanding of the distinct properties exhibited by shallow and deep layers. Except for Figure 1, "a shallow layer" denotes the first layer within a stage of ResNets, and "deep layers" refers to the rest of layers within this stage. The following analyses in Section 2 indicate that the layers within the same stage exhibit similar patterns to the layers throughout the entire model. Moreover, we have updated the paper and clarified these concepts in Section 2.
>
>
> > **W4.** Given the specific model heterogeneity settings under which InCo Aggregation is applicable, a discussion on how and where it isn't applicable would be useful.
>
> **Answer:**
> - Thank you for your feedback and suggestions. Currently, InCo Aggregation is applicable to CNNs (ResNets) and Vision Transformers (ViTs) architectures, but we have not yet explored its application to other network structures, such as LSTM. Furthermore, InCo Aggregation is primarily focused on addressing the issue of model heterogeneity within the same family of backbones with varying depths and widths. It is not suitable for handling model heterogeneity across completely different model families, such as a combination of CNNs and ViTs. This aspect requires further research. We have updated and discussed these limitations of InCo Aggregation in Appendix I.
> - However, regarding the problem of model heterogeneity across completely different architectures, we have identified two potential directions that could be explored in the context of InCo Aggregation. The first direction involves leveraging public datasets to perform knowledge distillation, where the knowledge from different model architectures is distilled onto one or several server models with the same architecture (more server models can store more knowledge). InCo Aggregation can then be performed on these server models to connect clients with different model architectures. This direction is similar to the approach taken in FedDF [1].
>
> [Our response continues.]

---

> ### Author Response · Authors · 2023-11-19
> **Response to Reviewer e8EV [2/4]**
>
> - The second method involves utilizing hypernetworks to generate parameters for corresponding layers. Different model architectures can be connected through hypernetworks, similar to how different input dimensions can be transformed into the same dimensions of layer parameters through hypernetworks. Similar to the previous direction, multiple server models can be generated using hypernetworks, followed by InCo Aggregation on the generated server models. Currently, there are several papers focusing on personalized federated learning that utilize hypernetworks for generating weights, such as [2], [3], and [4]. However, we have not yet come across any methods using hypernetworks for addressing model heterogeneity in federated learning.
> - These are preliminary ideas to make InCo Aggregation applicable to model heterogeneity across completely different backbones. However, further in-depth research and experimental analysis are required to explore these directions fully.
>
>
> Questions:
>
> > **Q1.** Could the authors define the notion of system level heterogeneity?
>
> **Answer:**
> - Thank you for your question. Please refer to the response regarding weakness 1.
>
> > **Q2.** Could the authors provide gradient plots like in Figs 2 and 3 for Stage 2's layers as well?
>
> **Answer:**
> - Thank you for bringing this issue to our attention. We have updated the results for stage 2 in Appendix G.4. In contrast to the gradient distributions of stage 3, the differences in gradient distributions across different layers are less evident for stage 2. This can be observed from Figure 1a, where the CKA similarity for stage 2 is considerably higher than that of stage 3. The higher similarity indicates that stage 2 is relatively less biased and more generalized compared to stage 3, resulting in less noticeable differences in gradient distributions. This observation further supports the relationship between similarity and smoothness, as higher similarity leads to smoother distributions.
>
> > **Q3.** Could the authors provide an explicit definition of which layers can be considered deep vs.shallow?
>
> **Answer:**
> - Thank you for your question. Please refer to the response regarding weakness 3. We have updated the paper and clarified these concepts in Section 2.
>
> > **Q4.** Could the authors provide an explanation of whether this pattern carries over to ViTs as well? Could the authors discuss further about similarity patterns in ViT's and how this impact the observations and InCo aggregation as a whole?
>
> **Answer:**
> - Thank you for your comments. We have updated the cross-environment similarity for ViTs in Appendix G.3, along with the analysis of their gradient distributions and the relationships with similarity.
> - Similar to the gradient analyses conducted for ResNets, we have performed the analysis of gradient distributions for ViTs. In our investigation, we have analyzed the outputs from the norm1 and norm2 layers within the ViT blocks and have also applied InCo Aggregation to these layers. The selection of norm1 and norm2 layers is motivated by the significance of Layer Norm in the architecture of transformers [5]. Additionally, we have chosen Block7 and Block11 for analysis as, in the context of heterogeneous models, Block7 is the final layer of the smallest ViTs, while Block11 represents the final layer of the largest ViTs.
> - From Figure 12a and 12b, we observe that the cross-environment similarities derived from the shallow layer norm (norm1) are higher compared to those from the deep layer norm (norm2). Moreover, similar to the analysis conducted for ResNets, we discover that the distributions of norm1 in ViTs exhibit greater smoothness compared to norm2, as depicted in Figure 13 and Figure 14. These findings reinforce the notion that InCo Aggregation is indeed suitable for ViTs.
>
> > **Q5.** Could the authors provide more detail explanations for the exact setups used to generate Fig. 1, 2, and 3?
>
> **Answer:**
> - Thank you for your comments. Due to page limitations, we have put the detailed experimental setups for the case study in Appendix F.1. The comprehensive experimental settings are provided below.
>
> [Our response continues.]

---

> > ### Author Response · Authors · 2023-11-19
> > **Response to Reviewer e8EV [3/4]**
> >
> > - In the case study, we have five ResNet models which are stage splitting from ResNet26, resulting in ResNet10, ResNet14, ResNet18, ResNet22, and ResNet26. Five ViTs models are ViT-S/8, ViT-S/9, ViT-S/10, ViT-S/11, ViT-S/12, the results from the layer splitting of ViT-S/12. The model prototypes are the same as the experiment settings. To quantify a model's degree of bias towards its local dataset, we use CKA similarities among the clients based on the outputs from the same stages in ResNet (ResNets of different sizes always contain four stages) and the outputs from the same layers in Vision Transformers (ViTs). Specifically, we measure the averaged CKA similarities according to the outputs from the same batch of test data. The range of CKA is between 0 and 1, and a higher CKA score means more similar paired features. We train FedAvg under three settings: IID with the homogeneous setting, Non-IID with the homogeneous setting, and Non-IID with the heterogeneous setting. Non-IID is defined by $\alpha=0.5$ from Dirichlet distribution. FedAvg only aggregates gradients from the models sharing the same architectures under the heterogeneous model setting.
> > - For ResNets, we conduct training 100 communication rounds, while only 20 rounds for ViTs. The local training epochs for clients are five for all settings. We use Adam optimizer with default parameter settings for all client models (ResNets and ViTs), and the batch size is 64.
> > - We use two small federated scales. One is ten clients deployed the same model architecture (ResNet18 for ResNets and ViT-S/12 for ViTs), which is called a homogeneous setting. The other is ten clients with five different model architectures, which is a heterogeneous setting. This setting means that we have five groups whose architectures are heterogeneous, but the clients belonging to the same group have the same architectures.
> >
> > > **Q6.** Could the authors provide more insight into how Fig. 3 would vary when tested across multiple trials?
> >
> > **Answer:**
> > - Thank you for pointing out this issue. We have updated a new Figure 3 with different random seed in Appendix G.4, specifically in Figure 15c and 15d. The updated results demonstrate that the distribution of stage3 conv0 remains smoother compared to conv1.
> > - Moreover, we believe that Figure 3 will display variations in the peak values of the distributions across multiple trials, as depicted in Figure 15d. For instance, in Figure 3b, the peak value may vary and potentially appear on the right side. It is important to note that the peak values may not necessarily follow a consistent pattern, where one side exhibits a significantly higher peak while the other side has a lower peak. In fact, both peaks may be relatively similar in height, similar to the observations for ViTs shown in Figure 14d. However, one consistent result is that the distribution of stage3 conv1 remains non-smooth, while the distribution of stage3 conv0 is smoother compared to conv1.
> >
> > > **Q7.** Could the authors provide the standard deviation values for InCo-based methods?
> >
> > - Thank you for your suggestion. We have updated the standard deviation values for InCo-based methods in all experiments in Table 5 and Table 6. Specifically, in the experiments involving model-heterogeneous methods, the standard deviation values for InCo-based methods are presented as follows:
> >
> > |      | FashionMNIST-0.5 | FashionMNIST-1.0 | SVHN-0.5 | SVHN-1.0 | CIFAR10-0.5 | CIFAR10-1.0 |
> > | ---- | ---------------- | ---------------- | -------- | -------- | ----------- | ----------- |
> > | HeteroFL+InCo | 90.0 $\pm$ 1.2  | 90.4 $\pm$ 1.1  | 92.1 $\pm$ 1.0  | 93.5 $\pm$ 1.5  | 68.2 $\pm$ 3.8 | 71.2 $\pm$ 3.4  |
> > | InclusiveFL+InCo | 90.1 $\pm$ 1.5  | 90.5 $\pm$ 1.3  | 90.6 $\pm$ 1.7  | 90.9 $\pm$ 1.9  | 69.1 $\pm$ 2.8 | 72.3 $\pm$ 3.1  |
> > | FedRolex+InCo | 90.4 $\pm$ 1.4  | 91.3 $\pm$ 1.1  | 92.8 $\pm$ 1.5  | 93.4 $\pm$ 1.6  | 67.9 $\pm$ 2.9 | 75.6 $\pm$ 2.6  |
> > | ScaleFL+InCo | 91.5 $\pm$ 1.0  | 91.7 $\pm$ 1.1  | 93.4 $\pm$ 0.9  | 93.6 $\pm$ 0.9  | 73.8 $\pm$ 3.2 | 76.1 $\pm$ 2.6  |
> >
> > [Our response continues.]

---

> > > ### Author Response · Authors · 2023-11-19
> > > **Response to Reviewer e8EV [4/4]**
> > >
> > > > **Q8.** Could the authors provide a detailed breakdown of how the experimental settings differ from the original works?
> > >
> > > **Answer:**
> > > - Thank you for your comment. The biggest difference leading to variations in results lies in the different model architectures. In the ScaleFL paper, the server model is based on ResNet110, while in the FedRolex paper, the server model is based on ResNet18. In our experiments, we used ResNet26, with the layer parameters obtained from the PyTorch source code. In the PyTorch source code, the kernel_size, stride, and padding of the first convolutional layer are set to [7, 2, 3], while in the source code of FedRolex and ScaleFL, the corresponding parameters are [3, 1, 1] and [5, 2, 3], respectively. To maintain consistency in the overall experimental results, we adopted the settings from the PyTorch source code ([7, 2, 3]) for both FedRolex and ScaleFL, ensuring that they can reach a converged state under this configuration.
> > > - The Non-IID setting of the datasets also differs. In FedRolex, the experiments were conducted with each client having 2 or 5 labels. In ScaleFL, the dataset was generated using a Dirichlet distribution with $\alpha=1$ and $\alpha=100$. In our experiments, we set the dataset using a Dirichlet distribution with $\alpha=0.5$ and $\alpha=1$. For all ResNet models, we set the number of global rounds to 500 and the number of client training rounds to 5. With these experimental parameter settings, we can ensure that both FedRolex and ScaleFL have converged.
> > > - Other settings, such as the learning rate for the optimizer and the step size for the learning rate scheduler, were kept consistent with the corresponding paper's source code.
> > >
> > >
> > > > **Q9.** A discussion on how and where it isn't applicable is critical to understand and apply the proposed method.
> > >
> > > **Answer:**
> > > - Thank you for your comment. Please refer to the response regarding weakness 4.
> > >
> > >
> > > [1] Ensemble distillation for robust model fusion in federated learning. NeurIPS 2020.
> > >
> > > [2] Personalized federated learning using hypernetworks. ICML 2021.
> > >
> > > [3] Layer-wised Model Aggregation for Personalized Federated Learning. CVPR 2022.
> > >
> > > [4] Efficient model personalization in federated learning via client-specific prompt generation. ICCV 2023.
> > >
> > > [5] On layer normalization in the transformer architecture. ICML 2020.

---

> > > > ### Author Response · Authors · 2023-11-22
> > > > **Official Comment by Authors**
> > > >
> > > > Dear Reviewer e8EV,
> > > >
> > > > We sincerely thank you for investing your time in reviewing our work and for providing valuable feedback. Your comments have been truly invaluable in enhancing the clarity and readability of our content. As we approach the end of the discussion period, please feel free to reach out if you have any additional questions or require further information. We are here to assist you with great enthusiasm. If our responses have met your expectations, we would greatly appreciate your acknowledgment of our efforts. Once again, we express our gratitude for your significant contribution to improving our work. We eagerly anticipate receiving your feedback.
> > > >
> > > > Best regards,
> > > >
> > > > The authors

---

> > > > > ### Comment · Reviewer_e8EV · 2023-12-02
> > > > > **Response to Authors**
> > > > >
> > > > > I commend and greatly appreciate the detailed responses (and revisions) provided for each of the weaknesses/questions from the original review. They offer much need clarity and insight into the method and experimental setting needed  grasp the importance of the contributions.
> > > > > I believe that adding the necessary nuance, of where and how InCo in applicable, is critical in framing its use-case, thus quickly bridging the problem statement with the solution.
> > > > > In light of the responses from the authors, I have positively updated my review score.

---

### Official Review · Reviewer_vLVW · 2023-11-01

**Soundness:** 3 good
**Presentation:** 4 excellent
**Contribution:** 3 good
**Rating:** 8
**Confidence:** 3

**Summary:**

The authors observe that layer similarities are related to accuracy for some FL models. They also observe that layer similarities are related to gradient distribution smoothness. Based on these observations, they modify the learning approach to increase similarities between shallow and deep layer gradients. This modification results in improved accuracy for a variety of FL methods and datasets. They also provide for layer splitting and other engineering necessities to evaluate their ideas, but I found the observations and modification above to be the most interesting and novel.

**Strengths:**

+ This is an interesting paper to read.

+ The paper combines experimental approaches to scientific discovery and system engineering to use these discoveries to produce improved accuracy.

+ The visualizations of findings and diagrams are clear.

+ The writing is well organized.

**Weaknesses:**

- The reasons for the relationships observed by the authors are not well explained, although Section 5.4 makes a mostly unsuccessful attempt at doing so. The authors find that high-accuracy models tend to have particular properties and push the learning process to produce those properties without well explaining why the properties result in accuracy. They do, however, demonstrate that their approach works so it's a question of depth of understanding, not merit.

- Section 3.3 seems central to enabling improvement but it is relatively short, without much justification for design decisions. It states what is done but now why this is the most appropriate approach.

**Questions:**

1) Is there a fundamental justification for the form of expression 1, or might other expressions perform as well or better?

2) Why did you decide to simply constrain gopt from opposing g0 (with an inequality) instead of imposing a cost that increases with decreasing dot product?

3) What is the relationship between your findings and those regarding the contribution of residual connections in ResNets?

**Details Of Ethics Concerns:**

When searching for related work, I found a paper with the same title and content that list the author names. Is this consistent with the double-blind review rules?

EDIT: The authors gave an explanation that appears to show compliance with conference policies. The conference policy is self-defeating, but the authors shouldn't suffer as a result of that.

---

> ### Author Response · Authors · 2023-11-19
> **Response to Reviewer e8EV [1/3]**
>
> Thank you for your valuable and constructive review. We now address your main concerns as follows.
>
>  Weaknesses:
>
> > **W1.** The reasons for the relationships observed by the authors are not well explained.
>
> **Answer:**
> - Thank you for your feedback regarding this matter. First, we introduce the reasons why high CKA similarities imply better model performance. High CKA similarity suggests a high degree of alignment between the feature spaces of different models. This can be validated by looking at the eigendecomposition in the calculation of CKA similarity [1].
> Suppose $X\in \mathbb{R}^{n\times p}, Y\in \mathbb{R}^{n\times q}$ are centered feature vectors (i.e., with column mean 0).
> $$
>     \begin{align} \text{CKA}_{\text{linear}}(X,Y)&=\text{CKA}(XX^\text{T},YY^\text{T}) \\\\
> &=\frac{\text{tr}(XX^\text{T}YY^\text{T})}{\sqrt{\text{tr}(XX^\text{T}XX^\text{T})\text{tr}(YY^\text{T}YY^\text{T})}} \\\\
> &=\frac{\sum\_{i=1}^p\sum\_{j=1}^q\alpha_i\beta_j\langle\mathbf{u}_i,\mathbf{v}_j\rangle^2}{\sqrt{\sum\_{i=1}^p\alpha\_i^2}\sqrt{\sum\_{i=1}^q\beta\_i^2}},
> \end{align}
> $$
> where $\{\mathbf{u}_i\}\_{i=1}^n, \{\mathbf{v}_i\}\_{i=1}^n$ are the left-singular vectors of $X,Y$, and $\alpha_j,\beta_j$ are the $j$-th largest eigenvalues of $XX^\text{T}, YY^\text{T}$ (squared singular value of $X,Y$). Hence, by the weighted inner product of the eigenvectors, CKA tells the amount of variance in X that Y explains (and vice versa), suggesting how similarly the two representations of a data point lie in the space. Then, consider two local models $f(x;\theta_1),f(x;\theta_2)$, parameterized by $\theta_1,\theta_2$ repectively, who update themselves locally with data $X_1, X_2$. High CKA similarity between $f(x,\theta_1)$ and $f(x,\theta_2)$ means that for $x^*\in X_1\setminus X_2$, it holds that $f(x^*;\theta_2)$ still approximates $f(x^*;\theta_1)$ even if $x^*$ is unseen by $f(x;\theta_2)$ and $X_2\setminus X_1$ contibutes to $\theta_2$'s update. Therefore, InCo Aggregation successfully improves the generalizability of local models and eventually achieves better performance.
> - The discussion above indicates that lower CKA similarity reflects lower similarity in the features generated by the models, resulting in increased bias. These properties highlight a key issue, namely the bias introduced by client models in a model heterogeneity setting, which can stem from variations in model architectures. Therefore, in Section 5.4, we address the bias issue and demonstrate how InCo aggregation achieves higher accuracy by mitigating this bias.
> - In Section 5.4, Figures 9d-9f provide vision representations of how InCoAvg's features exhibit a higher degree of generalization compared to FedAvg and HeteroAvg, which often display biased phenomena stemming from model heterogeneity. Furthermore, Figure 10a quantitatively describes the efficacy of InCoAvg in promoting model generalization using the CKA similarity measure. Specifically, InCoAvg exhibits higher CKA similarity in deep layers compared to FedAvg and HeteroAvg. The results obtained from Figures 9d-9f and 10a validate that the utilization of internal cross-layer gradients effectively facilitates the transfer of generalized knowledge from shallow layers to deep layers. This transfer mechanism helps alleviate the bias issue within models, consequently improving their overall performance.
>
> > **W2.** Section 3.3 seems central to enabling improvement but it is relatively short.
>
> **Answer:**
> - Thank you for bringing up this issue. We introduce the inequality constraint to limit the influence of $g_0$ on $g_{opt}$ because we want to ensure that $g_{opt}$ is primarily driven by the original gradients of the current layer, denoted as $g_k$. Our objective is to find the optimal projective gradients, denoted as $g_{opt}$, which strike a balance between being as close as possible to $g_k$ while maintaining alignment with $g_0$. This alignment ensures that $g_k$ is not hindered by the influence of $g_0$ while allowing $g_{opt}$ to acquire the beneficial knowledge for $g_k$ from $g_0$. In other words, we aim for $g_{opt}$ to capture the advantageous information contained within $g_0$ without impeding the progress of $g_k$. Hence, we employ this simple yet effective constraint. We have added an explanation of this aspect in Section 3.3.
>
>
> Questions:
>
> > **Q1 & Q2.** Is there a fundamental justification for the form of expression 1? Why did you decide to simply constrain gopt from opposing g0 (with an inequality)?
>
> **Answer:**
>
> - Thank you for your concern. We can show that constraining $g_{opt}$ from opposing $g_0$ (with an inequality) is better than imposing a cost that increases with decreasing dot product, as supported by the following theoretical analysis.
> - Denote constraining $g_{opt}$ from opposing $g_0$ (with an inequality) as $L_1$, i.e., $L_1=||g_k-g_{opt}||\_2^2,\,s.t.\,\langle g_{opt},g_0\rangle\geq0$.
>
> [Our response continues.]

---

> ### Author Response · Authors · 2023-11-19
> **Response to Reviewer e8EV [2/3]**
>
> - Denote imposing a cost that increases with decreasing dot product as $L_2$, i.e., $L_2=||g_k-g_{opt}||\_2^2-\lambda \langle g_{opt},g_0\rangle$ for a fixed $\lambda\geq0$.
> - Then, we can show that $\forall \lambda\geq0, \min L_1\geq \min L_2$ as follows, which means that $\min L_1$ always give an upperbound for $\min L_2$ no matter what the value of $\lambda$ is. As long as we ensure that $L_1$ is small enough, then $L_2$ is automatically smaller. We abbreviate $g_{opt}$ as $g$.
> \begin{equation}
> \begin{aligned}
> L_1&\geq\min_{\substack{g\text{ s.t. }\langle g,g_0\rangle\geq0}} L_1 \\\\
> &\geq \min_{\substack{g\text{ s.t. }\langle g,g_0\rangle\geq0}} ||g_{k}-g||\_{2}^{2} - \lambda \langle g,g_0\rangle,\text{ for a fixed }\lambda\geq0 \\\\
> &\geq \min_{\substack{g\text{ unconstrained}}} ||g_{k}-g||\_{2}^{2} - \lambda \langle g,g_0\rangle,\text{ for a fixed }\lambda\geq0 \\\\
> &= \min_g L_2
> \end{aligned}
> \end{equation}
> The second inequality follows from $\lambda \langle g,g_0\rangle\geq0$. The third inequality follows from relaxing the constraints of $g$. Specifically, given the inequality constraint $\min L_1 \geq \min L_2$, if we aim to satisfy $\epsilon \geq \min L_2$, where $\epsilon$ is a suitably small value, it follows that $\epsilon \geq \min L_1 \geq \min L_2$. Thus, we can establish the equivalence between $L_1$ and $L_2$ under these conditions.
> - Moreover, the second point is that the form of Expression 1 enables us to determine the value of the leveraging weight $\lambda$ in the Lagrangian without requiring intricate manual parameter tuning. In fact, the form of $L_2$ corresponds to the form of the Lagrangian of $\min L_1$, i.e.,
> \begin{equation}
> \begin{aligned}
> L(g, \lambda)=||g_{k}-g||_{2}^{2} - \lambda g^Tg_0, \\\\
> L_2(g)=||g_k-g||_2^2-\lambda g^Tg_0.
> \end{aligned}
> \end{equation}
> - In this case, the solution can be derived as $g=g_k+\lambda g_0/2$. If we use other form, such as $L_2$, the value of $\lambda$ would become a hyperparameter requiring manual adjustment. However, by utilizing the presented inequality constraint in the paper, we can solve for $\lambda=-\frac{2b}{a}$ using the Lagrange dual function, where $a=(g_0)^Tg_0$ and $b=g_k^Tg_0$. This particular approach enables dynamic adjustment of the $\lambda$ value based on varying gradients, eliminating the need for manual tuning. The more details about the Lagrangian are explained in Appendix C.
>
>
>
> > **Q3.** What is the relationship between your findings and those regarding the contribution of residual connections in ResNets?
>
> **Answer:**
>
> - Thank you for pointing out this issue. The goal of residual connections is to avoid exploding and vanishing gradients to facilitate the training of a single model [2], while cross-layer gradients aim to increase the layer similarities across a group of models that are jointly optimized in federated learning. Specifically, residual connections modify forward passes by adding the shallow-layer outputs to those of the deep layers. In contrast, cross-layer gradients operate on the gradients calculated by back-propagation. We present the distinct gradient outcomes of the two methods in the following.
> - To illustrate the differences in gradient updates between these two methods, let's consider a simple example involving three consecutive layers of a feedforward neural network, indexed as $i-1, i, i+1$. We will use the notation $f(\cdot;W_k)$ to represent the calculations in the $k$-th layer. Given the input $\mathbf{x_{i-2}}$ to layer $i-1$, the output of the previous layer serves as the input to the next layer, resulting in sequential outputs $\mathbf{x_{i-1}}, \mathbf{x_{i}}, \mathbf{x_{i+1}}$.
> $$
> \begin{align}
> \mathbf{x_{i-1}}&=f(\mathbf{x_{i-2}}; W_{i-1}),\\\\
> \mathbf{x_{i}}&=f(\mathbf{x_{i-1}}; W_{i}),\\\\
> \mathbf{x_{i+1}}&=f(\mathbf{x_{i}}; W_{i+1}).
> \end{align}
> $$
> - As illustrated in Figure 11 in Appendix B, there is an additional operation that directs $\mathbf{x_{i}}$ to $\mathbf{x_{i+1}}$, formulated as $\mathbf{x_{i+1}'}=\mathbf{x_{i+1}}+\mathbf{x_{i}}$ in the case of residual connections. The gradient of $W_{i}$ is
> \begin{aligned}
>         g_{W_i}=\frac{\partial loss}{\partial(\mathbf{x_{i+1}}+\mathbf{x_{i}})}
>         \cdot \left(\frac{\partial \mathbf{x_{i+1}}}{\partial W_i}+\frac{\partial \mathbf{x_{i}}}{\partial W_i}\right).
>     \end{aligned}
> - In the case of cross-layer gradients, the gradient of $W_{i}$ is
> \begin{aligned}
>         g_{W_i}=\frac{\partial loss}{\partial \mathbf{x}_{i+1}}\cdot \left( \frac{\partial \mathbf{x\_{i+1}}}{\partial W\_i}+\frac{\partial\mathbf{x\_{i+1}}}{\partial W\_{i-1}}\right).
>     \end{aligned}
> - Hence, it becomes evident that distinctions emerge between our proposed method and residual connections employed in ResNets, as demonstrated through this example. We further elaborate this issue with more comprehensive insights in Appendix B.
>
> [Our response continues.]

---

> > ### Author Response · Authors · 2023-11-19
> > **Response to Reviewer e8EV [3/3]**
> >
> > > The issue of Ethics Review
> > - According to the Dual Submission Policy outlined in ICLR2024, it is not in violation of the double-blind review rules to upload the paper to arXiv. The policy provides the following description regarding this matter in https://iclr.cc/Conferences/2024/CallForPapers : "The policy is enforced during the whole reviewing process period. Submission of the paper to archival repositories such as arXiv is allowed during the review period."
> >
> > [1] Similarity of neural network representations revisited. ICML 2019.
> >
> > [2] Identity mappings in deep residual networks. ECCV 2016.

---

> > > ### Comment · Reviewer_vLVW · 2023-11-22
> > > **Understood**
> > >
> > > The policy does appear to permit revealing authorship in this manner. It's a rather odd policy, because it defeats the purpose of blind review, i.e., any authors who want to reveal their identities to the reviewers can do so by concurrently releasing the paper under review on arXiv. Competent reviewers are very likely to find that paper when reviewing. However, that is a problem with the conference policy and not with the authors of this paper, so I don't think the authors should suffer any negative consequences in this case.

---

> > ### Comment · Reviewer_vLVW · 2023-11-22
> > **Thank you**
> >
> > Thank you for all your other clarifications. They are quite helpful.

---

> > > ### Author Response · Authors · 2023-11-23
> > > **Official Comment by Authors**
> > >
> > > Thank you for your acknowledgment of our efforts. We are glad to hear that the clarifications are helpful to you.

---

### Author Response · Authors · 2023-11-19
**Key modifications in the updated version.**

Dear reviewers and AC,

We sincerely thank all the reviewers for their constructive and valuable comments, which have greatly helped us in making substantial improvements to our paper. We would like to highlight the key modifications made for clarity and convenience:

- In the Introduction (Section 1), we refine and expand the descriptions and definitions of system heterogeneity and model heterogeneity to provide a more comprehensive understanding of the problem domain.
- Figure 6 is updated to include a contour plot of the gradients and clearer indications of the client and global optima.
- In the Gradients Divergence Alleviation (Section 3.3), we provide an updated motivation for incorporating the inequality constraint in Equation 1.
- For Table 3 in the experiments section, we update a comparison of communication overhead and computational resource utilization (FLOPs), and we also discuss these results in Section 5.2.
- In Appendix A, we add a detailed introduction for CKA similarity.
- In Appendix F, we add a detailed problem formulation for federated learning with model heterogeneity.
- In Appendix G.3, we update a gradient analysis for ViTs, contributing to a more thorough exploration of applicability of InCo Aggregation.
- In Appendix I, we discuss limitations and future directions for InCo Aggregation.

Moreover, the updated parts are highlighted in red. We combine the main context and Appendix as one PDF for convenience. At last, we are grateful for the time and effort put in by the reviewers in evaluating this work. We are looking forward to engaging with the reviewers.

Sincerely,

The authors

---

### Meta-Review · Area_Chair_ipQo · 2023-12-06

**Metareview:**

This paper addresses the challenge of heterogeneous models in federated learning, uncovering a key insight: shallow layers tend to have similar gradient distributions, unlike the varied distributions in deeper layers. This observation is crucial, as higher gradient similarity is linked to improved accuracy. With this solid motivation, the paper introduces Internal Cross-layer gradients (InCo) Aggregation, which leverages normalized cross-layer gradients to improve the similarity across clients in deeper layers, without additional communication between clients. The paper conducts comprehensive experiments and thoroughly shows the proposed approach consistently improves all baseline approaches.

During the discussion phase, the authors presented detailed and comprehensive replies to address reviewers' concerns, and most of the reviewers supported the acceptance except reviewer ibVy, who did not respond to the rebuttal. AC thinks that the authors' reply would be a sufficient answer to the reviewer ibVy's questions and concerns. Overall, AC is happy to recommend the acceptance.

**Justification For Why Not Higher Score:**

Too many parts of the submitted draft were updated during the rebuttal period.

**Justification For Why Not Lower Score:**

All reviewers strongly support this paper except reviewer ibVy, who did not actively participate in the discussion.

---

### Decision · Program_Chairs · 2024-01-16

Accept (poster)